# Plots unlock time-series understanding in multimodal models

## Abstract

While multimodal foundation models can now natively work with data beyond text, they remain underutilized in analyzing the considerable amounts of multi-dimensional time-series data in fields like healthcare, finance, and social sciences, representing a missed opportunity for richer, data-driven insights. This paper proposes a simple but effective method that leverages the existing vision encoders of these models to "see" time-series data via plots, avoiding the need for additional, potentially costly, model training. Our empirical evaluations show that this approach outperforms providing the raw time-series data as text, with the additional benefit that visual time-series representations demonstrate up to a 90% reduction in model API costs. We validate our hypothesis through synthetic data tasks of increasing complexity, progressing from simple functional form identification on clean data, to extracting trends from noisy scatter plots. To demonstrate generalizability from synthetic tasks with clear reasoning steps to more complex, real-world scenarios, we apply our approach to consumer health tasks – specifically fall detection, activity recognition, and readiness assessment – which involve heterogeneous, noisy data and multi-step reasoning. The overall success in plot performance over text performance (up to an 120% performance increase on zero-shot synthetic tasks, and up to 150% performance increase on real-world tasks), across both GPT and Gemini model families, highlights our approach's potential for making the best use of the native capabilities of foundation models.

## 1 Introduction

Multimodal models like GPT4 (Achiam et al., 2023) and Gemini (Gemini Team et al., 2023) are trained to understand visual information natively. However, they are not specifically trained to understand time-series data – in particular, the tokenizers for large language models (LLMs) are not well-suited for representing large sequences of floating point numbers (Spathis & Kawsar, 2024). This mirrors the human approach (Card et al., 1999; Yalçin et al., 2016); we cannot easily make sense of a long array of floating point numbers - instead our first instinct is often to visualize the data through plotting, followed by extracting insights through statistical analysis.

We investigate the hypothesis that multimodal models understand time-series data better through their vision encoders than through the textual representation of the sequences using synthetic and real-world data experiments. Our synthetic data experiments allow us to closely control the difficulty of tasks through the addition of noise and by changing the number of points in each function. We also use a mix of tasks that require a differing number of reasoning steps, as well as different kinds of reasoning, to get a correct answer.

Fall detection and activity recognition are both real-world tasks that make use of inertial measurement units (IMUs) from mobile phones or wearable devices. IMUs are 6-dimensional waveforms consisting of 3 axes of acceleration data and 3 axes of angular velocity data. The fall detection task consists of classifying an IMU waveform segment into one of three classes: Fall, Active Daily Living (ADL) or Near Fall (a hard negative class). The activity recognition task consists of classifying a waveform segment into one of five classes: Sitting, Standing, Walking, Cycling or Stairs.

By contrast, the readiness assessment task is a binary classification of 28 days of training load data from a single user into a state of undertraining or overtraining. Because of the tabular nature of the

data, the plot version is presented as a bar plot. This is not the ideal setting for our method - we believe it's best used when the amount of data exceeds what's reasonably presentable in a text table.

Our findings show that when using our plot-based approach multimodal models perform much better on tasks where the result is dependent on understanding the overall trend. We find specific examples of this when identifying the functional form, the number of clusters, the correlation between two functions, and on the real-world pattern-recognition tasks of activity recognition and fall detection. For example, GPT4o using plots on the functional form identification task shows a performance improvement of 122% over using the text representation. On other tasks that require more advanced reasoning such as multi-step or connecting trend shapes with sequence magnitudes (e.g. identifying derivatives), and on tasks with tabular data (e.g. readiness assessment), the performance is equivalent. However, there is a substantial cost difference between vision and text prompts, which is particularly pronounced on very long-context tasks, as the same information in a long sequence that requires many (10,000's to 100,000's) text tokens can be represented in one plot with many fewer (100's to 1000's) vision tokens. While vision tokens are more expensive than text tokens, the difference in unit cost is much lower than the orders of magnitude difference in overall prompt length, so that the total cost is still much lower using the vision approach. This difference is especially relevant on tasks where extensive few-shots are required to achieve good performance, and optimizing token efficiency translates to significant resource savings. Not only does this plot-based approach achieve better performance while being more efficient, it is also completely generalizable across any task that involves reasoning about a long, complex time-series as it requires zero additional model training.

Our work empirically evaluates the relative performance of the native capabilities of existing multimodal foundation models on visual versus textual representations of time-series data. This contribution furthers the understanding of modern foundation models, which is important in real-world contexts as user-facing products continue to develop multimodal sophistication and users interrogate data types that are more complex than can be easily represented with text only. While we do not claim to achieve the same absolute performance as models trained for specific tasks, and likely our approach will not match such models, our results presented here nonetheless show that in contexts when one relies on a foundation model to ingest any general time-series data with *a priori* unknown characteristics, a visual representation is on balance likely to yield better, and cheaper, results.

## 2 RELATED WORK

**Forecasting** We use the term "time-series understanding" in this paper to distinguish from time-series forecasting. Time-series forecasting predicts future data points based on points seen so far, whereas we are primarily interested in the setting where we connect the time-series data to a multimodal model for further analysis. In particular, we want to show that multimodal models can reason about overall trends, the relationship between multiple time-series, overall clustering of data, and other time-series understanding tasks. Forecasting has been a very productive area of the field – for a closer look at the literature, the survey paper by Zhang et al. (2024) tracks a wide range of time-series understanding and forecasting work.

**Time-series models** There are several existing approaches that train time-series encoders for specific tasks or domains. For example, Chan et al. (2024) trained a domain-specific encoder for multimodal medical time-series analysis. Similarly, Cosentino et al. (2024) trained an encoder for the sleep data in their Digital Well-being task. In this paper, we are not claiming that our approach would outperform a task-specific model on a specific task – we claim that one can achieve much better performance from a foundation model by exploiting its native multimodal capabilities compared to using only text.

Others have also shown that training foundation models with Transformer architectures specifically to work in time-series contexts can lead to good results across tasks including mostly forecasting but also classification, anomaly detection and imputation. These trained models do especially well when the input time-series are carefully pre-processed and tokenized, including patching, scaling and quantization Das et al. (2023); Woo et al. (2024); Goswami et al. (2024); Ansari et al. (2024); Cai et al. (2023). While they did not train a new model, in LLMTime (Gruver et al., 2024) the authors showed that with careful tokenization, text-only LLMs can perform well at forecasting tasks;

we perform ablations based on their methods and select the best tokenizations accordingly for our text baselines.

In this work, our goal is to show that simply plotting the data without additional data preprocessing or model training is at the very least an easy first step, and might be a helpful approach when training a task-specific encoder from scratch may not be feasible due to the requirements on having additional paired data, compute and expertise.

**Vision models and visual representations** While studying multimodal models' abilities to reason about visual inputs, Rahmanzadehgervi et al. (2024) found that multimodal models are unable to reason effectively, although some follow-up work by Corin (2024) indicated that prompt engineering can fix losses. In any case, our results do not necessarily contradict this - for many of our tasks humans may be able to get perfect scores, and indeed the multimodal models do not. Regardless, our main claim that plots are better suited than text as input to a multimodal model for time-series understanding holds true.

Perhaps an inversion of our approach, DePlot (Liu et al., 2023) translated visual plots to numeric tables and operated on the tabular data. This approach may be sound for discrete data where the number of points remains small – cases where a human would be expected to understand a table of data well.

Closely related to our work are those methods that use vision-embedding models like Contrastive Language–Image Pre-training (CLIP) by Radford et al. (2021). Wimmer & Rekabsaz (2023) used CLIP to embed plots of financial time-series data, from which features are extracted for use by downstream classifiers. Since we use the vision encoders of the multimodal foundation models directly, there is no need for further feature extraction or downstream classifiers in our approach. IMU2CLIP (Moon et al., 2023) and ImageBind (Girdhar et al., 2023) used video and image data paired with waveforms to learn joint embeddings. Both of these works rely on existing paired waveform and video data to "bind" the modalities together, whereas we can simply plot the waveforms and use the existing multimodal vision encoder to derive our time-series embeddings.

**Measuring understanding** Past work has also investigated various approaches to measuring the degree to which models can understand and reason about various modalities of inputs, such as charts (CharXiv (Wang et al., 2024)), tables and figures (SPIQA (Pramanick et al., 2024)) and time-series themselves (TimeSeriesExam (Cai et al., 2024)). These works generally involve generating novel evaluation datasets, and in some cases (e.g. (Wang et al., 2024), (Pramanick et al., 2024)) rely on language models to generate the questions themselves. In our work we deliberately avoid this approach as it can introduce biases during evaluation that are hard to account for (e.g. favoring their own output as in (Panickssery et al., 2024)). In TimeSeriesExam (Cai et al., 2024), the authors also found, as we do, that models perform better on plot-based representations of time-series than the analogues text-based representations, though only demonstrate this on synthetic data as part of a carefully optimised exam generation algorithm.

## 3 METHODOLOGY

We evaluate our visual prompting method on both synthetic data and real-world use-cases. Synthetic data allows us to control the difficulty of the task by adding noise and altering the number of data points, and to investigate specific kinds of reasoning in isolation. We chose the synthetic tasks to align with the different steps of reasoning we hypothesize are required for the representative real-world use cases we test on. Note that in this context, "reasoning" refers to the high-level steps we believe humans would take to get to the right answer, rather than any formal modelling approach such as chain of thought. These tasks include understanding the local and global longitudinal signatures (trend and magnitude) of a time-series, and potentially comparing it with several other time-series (as in the case of multidimensional sensing). We summarize in Section 4.1 the different tasks in our experiments, along with the type of reasoning we are probing.

The goal of our work is to study specifically the difference in performance achieved by models when ingesting visual versus textual representations rather than the absolute performance on any one task with either modality. As such the appropriate baseline, and the one we use, is the models' performances on textual representations. We nonetheless include random choice baselines for con-

text to show there is utility in leveraging these models at all, and compare against state-of-the-art task-specific models (for two of the real-world tasks) for context.

## 3.1 STRUCTURED PROMPTING

We used the open-source structured prompting library Langfun (Peng, 2023) for all tasks in this paper, with the exception of the Readiness task (Section 4.2) which is processed in a privacy-preserving sandbox environment. The prompts and Langfun code snippets for all tasks (except Readiness) are provided in Appendix A.4 for reproducibility. The structured prompting approach in Langfun allows us to use target schemas for outputs, though we do not use the controlled generation feature (Gemini models) or structured output (GPT4o models), simply relying on the native formatting of the model to the correct schema.

## 3.2 MODELS

We tested all synthetic data tasks on two frontier models: Gemini Pro 1.5 (gemini-pro-001) and GPT4o (gpt4o-2024-08-06) and two smaller models Gemini Flash 1.5 (gemini-flash-001) and GPT4o-mini (gpt4o-mini-2024-07-18). We use a temperature of 0.1 for all our experiments, Supplementary Tables S33-S35 includes our ablations on temperature. All other sampling parameters remain at API defaults.

## 3.3 FLOATING POINT REPRESENTATION

In order to find the best textual representations, we ran ablations (Appendix A.3) inspired by LLM-Time (Gruver et al., 2024) on which floating point precision (2, 4, 8, 16) and separator (space or comma and space) to use. We also tested the scaling approach suggested by LLMTime. We found that the lower precision led to better performance. On synthetic tasks, the best performing separator differs per model, on Gemini we make use of the space separator whereas on the GPT4o family we use comma and space. On real-world tasks we make use of the space separator for all models as we did not observe a difference in performance on these tasks and the space separator uses fewer tokens.

## 3.4 STATISTICAL METHODS

For our aggregate results, we aggregate individual model responses to an overall performance quality metric (accuracy or mean absolute error (MAE)) over the task dimensions as described in Supplemental Section A.1.1. This produces multiple points from which we extract a distribution presented as a box-plot where the central line is the median, the edges of the boxes are the inter-quartile range (IQR), the whisker lengths extend to 1.5 times the IQR and outliers are presented as individual points. For our more detailed plots in Appendix A.2 we show 95% confidence intervals constructed from 1.96 times the standard error of the mean of the metric.

For real-world tasks, since we don't have the ability to regenerate the same problem, we instead make use of bootstrapping (with 1,000 replicates) to produce distributions of the macro-averaged $F_1$ scores from which we construct similar box-plots as for the synthetic tasks. Note that the distributions plotted in the real-world box-plots are thus expected to be tighter than the synthetic task plots, as they don't reflect independent replicates.

In Table 2, we present the median and IQRs of the differences between plot and text performances, with the difference taken such that a positive value always means the plot method performs better than the text method. For synthetic tasks, the median and IQR are calculated by directly creating the distribution of differences between the plot and text performances of different instances of the experiment, while for real-world tasks we create a distribution of differences by randomly sampling 1,000 random pairs of the bootstrapped metric distributions described immediately earlier.

For synthetic tasks only, we test for significant differences between the plot and text performances with a two-sided Wilcoxon signed-rank test (Wilcoxon, 1945). We apply a Bonferroni (Bland & Altman, 1995) correction for multiple comparisons within a task block. We could not apply the same hypothesis testing framework to the real-world tasks as the performance distributions were bootstrapped and thus not independent, violating the assumptions of the Wilcoxon test.

## 4 EXPERIMENTS AND RESULTS

| Type of data | Task | Reasoning | Time-series length |
|---|---|---|---|
| Synthetic | Functional form id. | Understanding of one overall trend | 10's - 1,000's |
| | Correlation of two lines | Understanding of two overall trends | 10's - 1,000's |
| | 2D cluster counting | Understanding and counting $N$ overall trends | 10's - 1,000's |
| | Derivative id. | Multi-step reasoning connecting two overall trends | 10's - 1,000's |
| | Quadratic derivative id. | Multi-step reasoning connecting two trends and magnitudes | 10's - 1,000's |
| Real-world | Fall detection from IMU data | Classify a pattern based on local spikes in multiple signals | 10,000's |
| | Activity recognition from IMU data | Classify a pattern based on global trends in multiple signals | 10,000's |
| | Readiness from wearable measures of training intensity | Compare a local trend with a global trend | 10's |

Table 1: Summary of the tasks we study in this paper, including the reasoning each requires and the length of the input time-series (based on the number of points). The "reasoning" column is a high-level summary of the steps needed to answer the tasks' questions correctly; tasks are detailed in Sections 4.1 and 4.2 and Supplementary Information Section A.1.

| Task (Metric) | Few-Shots | GPT4o-mini | Gemini Flash 1.5 | GPT4o | Gemini Pro 1.5 |
|---|---|---|---|---|---|
| Functional form id. (Accuracy) | 0 | **0.32** (0.18, 0.41)* | **0.22** (0.11, 0.40)* | **0.46** (0.31, 0.52)* | **0.04** (-0.08, 0.25) |
| Correlation of two lines (Accuracy) | 0 | **0.33** (0.17, 0.33)* | **0.25** (0.17, 0.33)* | **0.17** (0.00, 0.17)* | **0.33** (0.17, 0.50)* |
| 2D cluster counting (MAE) | 0 | **1.02** (0.53, 1.09)* | **1.67** (1.49, 1.80)* | **2.29** (1.84, 2.44)* | **1.82** (1.36, 2.06)* |
| Derivative id. (Accuracy) | 0 | **0.16** (0.12, 0.24)* | **0.08** (-0.04, 0.20) | 0.00 (-0.18, 0.12) | -0.02 (-0.16, 0.08) |
| Quadratic derivative id. (Accuracy) | 0 | **0.27** (0.23, 0.38)* | **0.15** (0.02, 0.30)* | -0.17 (-0.30, -0.10)* | **0.17** (-0.01, 0.24) |
| | 3 | **0.17** (0.12, 0.33)* | **0.32** (0.13, 0.34)* | **0.10** (-0.07, 0.17) | **0.28** (0.19, 0.43)* |
| Fall detection ($F_1$ score) | 1 | **0.03** (0.02, 0.05) | **0.11** (0.09, 0.12) | **0.32** (0.31, 0.34) | **0.13** (0.10, 0.15) |
| | 10 | **0.21** (0.19, 0.22) | **0.17** (0.15, 0.19) | **0.50** (0.49, 0.52) | **0.40** (0.38, 0.42) |
| Activity detection ($F_1$ score) | 1 | **0.09** (0.07, 0.11) | **0.12** (0.10, 0.14) | **0.03** (0.01, 0.06) | **0.20** (0.18, 0.22) |
| | 5 | **0.11** (0.09, 0.13) | **0.23** (0.21, 0.25) | **0.18** (0.15, 0.20) | **0.23** (0.21, 0.25) |
| Readiness ($F_1$ score) | 0 | – | -0.08 (-0.11, -0.06) | – | **0.07** (0.05, 0.09) |

Table 2: Summary of the experiments with 38 out of 42 results showing better performance on plots (bold numbers). Cells contain metric medians and IQRs. Stars in synthetic tasks only indicate statistically significant differences between plot and text metrics at 95% confidence corrected for multiple comparisons; we could not perform the same hypothesis testing on the real-world tasks. See Section 3.4 for statistical details and Supplementary Table S2 for relative differences between approaches.

## 4.1 SYNTHETIC DATA TASKS

Figure 1 summarizes all the zero-shot versions of the synthetic data tasks showing that plot-based methods outperform the text-based methods across GPT and Gemini model families, with few exceptions. Detailed task descriptions are provided in Supplementary Information Section A.1.1, and non-aggregated results over dataset parameters such as number of points and noise level are available in Supplementary Information Section A.2.

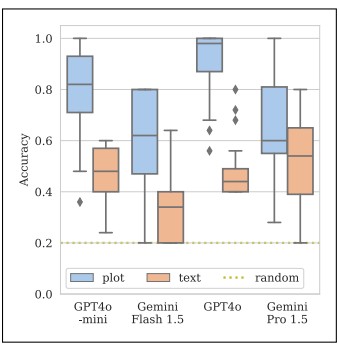 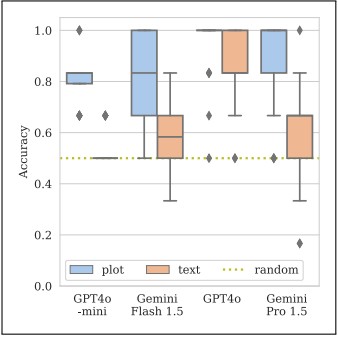 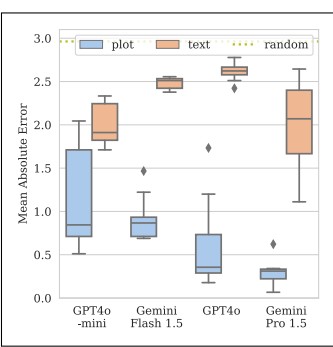

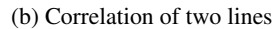 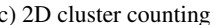

(a) Functional form id.    (b) Correlation of two lines    (c) 2D cluster counting

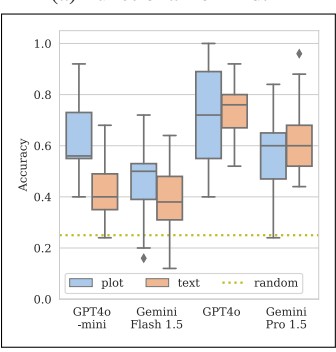 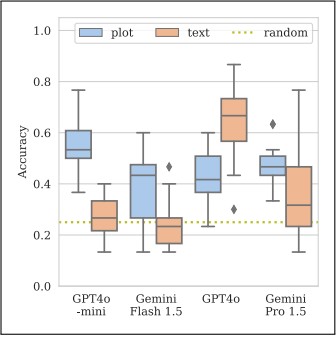

(d) Derivative id.    (e) Quadratic derivative id.

Figure 1: Zero-shot synthetic data results showing plot- and text-based accuracy (MAE for the cluster counting task) distributions for all models, with horizontal lines representing random performance. The results generally show better performance for plots compared to text across models.

**Functional form identification** (id.) This is the simplest task that requires only identifying one overall trend and correctly classifying it into one of five functional tasks (linear, quadratic, cubic, exponential or periodic). We generate a set number of points with a controlled level of noise according to one of the five function classes, and test the model's ability to label the global trend into the correct class.

**Correlation of two lines** This task now requires understanding the trends in two lines and comparing them against each other to correct identify whether the lines are positively or negatively correlated. Here we generate two linear series with controlled number of points and noise and predetermined slopes, measure the sign of the correlation analytically using the Pearson correlation coefficient, and probe the model's ability to identify the directional correlation.

**2D cluster counting** In this task, the model needs to correctly identify the $N$ number of clusters present in a set of points. To test this, we generate random points on a 2D grid with a set number of clusters and controlled minimum distance between cluster centers and standard deviation of the points about the centres. The model is then instructed to count the number of clusters.

**Derivative identification** This is a harder task: the model must now identify the correct first derivative (out of four choices) of the function provided in the question. The function and choices are presented as either plots or text. We pass various known functions to the model alongside four synthetic first derivatives, each corresponding to different functional classes, with controlled levels of

noise and number of points. The models are then asked to identify which of the multiple choices corresponds to the true derivative.

**Quadratic derivative identification** As a hard variant of derivative identification, we always pass a quadratic function and present four different linear functions as multiple choice answers, with controlled levels of noise and number of points. The model must now identify the correct linear function (out of four choices) that corresponds to the quadratic function's derivative, so it must pay attention to both the sign and magnitude of the linear slopes.

In order to investigate the quadratic derivative task further, we also ran experiments providing few-shot examples with reasoning traces with results shown in Figure 2. Here we find that GPT4o text zero-shot remains an outlier in its strong performance, but for the other models plot outperforms text, with few shots improving performance in the Gemini family for both plot and text, but reducing performance in the GPT family of models for both plot and text.

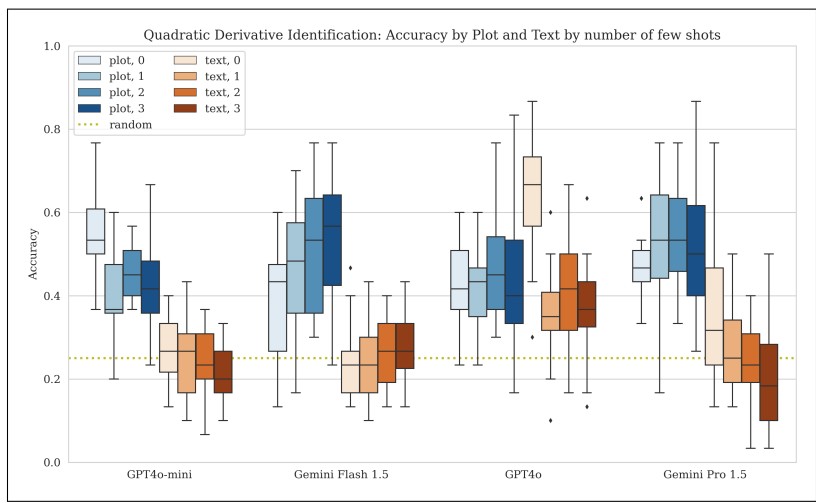

Figure 2: Quadratic derivative identification results show zero-shot plots outperform text, except for the outlier GPT4o model. When using few-shots, more examples generally improves the gain.

## 4.2 REAL-WORLD TASKS

Our synthetic tasks built up in complexity from understanding one trend globally (functional form identification) to understanding several trends simultaneously using global and local information (correlation and cluster counting) and complex multi-step reasoning (derivative identifications). We now probe tasks on real-world data that require a mix of these reasoning abilities, including simultaneous understanding of multiple sensor signals to uncover either local or global trends in the first two tasks (fall detection and activity recognition), and extracting two trends of different timescales in the last task (readiness).

**Fall detection from inertial measurement units (IMUs)** An IMU segment is a 6D-vector composed of 3-axes accelerometer signals and 3-axes gyroscope signals. The first real-world task we evaluate (results in Figure 3) is to classify whether a 15-second IMU segment recorded at 128hz contains a fall, a "near" fall or showed "active daily living" (ADL). The dataset used in the open-source IMU Fall Detection dataset (IMUFD, Aziz et al. (2017)).

Few-shot fall detection is a pattern-recognition task - typically a fall shows up on the IMU as a big spike in magnitude on multiple axes. What makes the task hard is the inclusion of the hard negative class of "Near" falls, where the participants of the study pretend to trip but recover before actually falling, creating similarly large changes in magnitude on the IMU.

**Activity recognition from IMUs** A further real-world IMU task we evaluate is to classify whether a 15-second IMU segment is one of five activity classes: "sit", "stand", "stairs", "walk" or "bike" (results in Figure 4). The dataset used in the open-source Heterogeneity Human Activity Recogni-

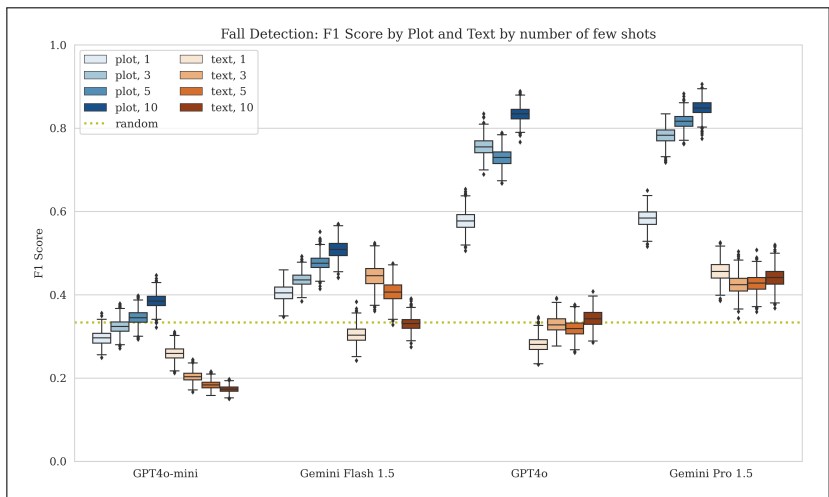

Figure 3: Results of fall detection task show consistently better plot performances across models and number of few-shots, with plot performance generally increasing with number of shots. The top plot models have 10-shot (sensitivity, specificity) as follows: Gemini Pro 1.5 - (0.84, 0.95) and GPT4o - (0.92, 0.81), compared to the state-of-the-art task-specific support-vector machine model reported by Aziz et al. (2017) which achieves (0.96, 0.96) (see Supplementary Table S1 for more details).

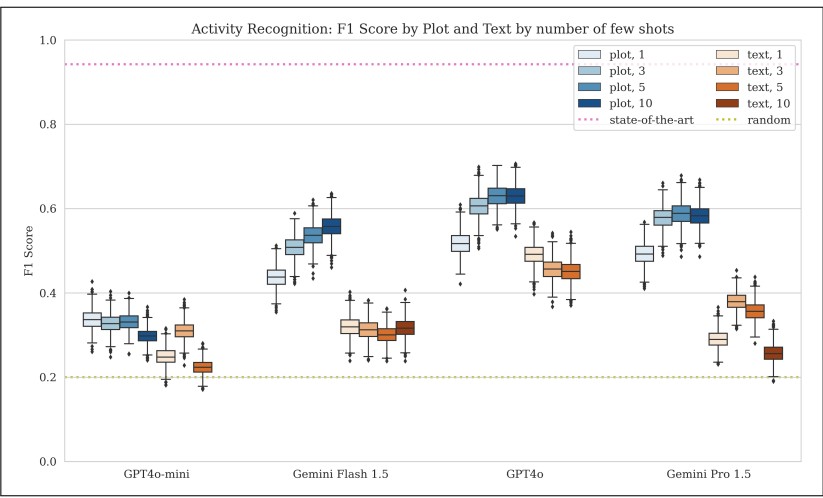

Figure 4: Results of activity detection task for all models across few-shot numbers (where context length allowed), showing overall improved performance for plots. The performance of the state-of-the-art deep-learning model reported by Kumar & Selvam (2022) is included for reference.

tion dataset (HHAR, Stisen et al. (2015)). As with Fall Detection we test performance with 1, 3, 5 and 10 few-shot examples, with 383 samples per model and number of few-shots. For this task the text representation of the 10 few-shot examples exceeded the GPT4o and GPT4o-mini 128k context windows, so these results are only shown for the Gemini models.

Activity recognition requires evaluating the entire IMU segment and correlating signals between different axes and sensors to determine the likely activity, as the noisy signals may only subtly change between "sit" and "stand" or "walk" and "stairs". The HHAR dataset was deliberately collected to be heterogeneous containing data collected from four different types of smartphone and two different types of smartwatch. The classes in the dataset aren't balanced so performance is reported using an $F_1$ score.

**Readiness** Estimating fitness readiness for a workout is a multicomponent task that involves assessment of health metrics, sleep, training load, and subjective feedback (Cosentino et al., 2024). Among those, training load analysis can be evaluated quantitatively and involves plot interpretation. Therefore, we framed the task as a binary classification problem (training load trending upwards or downwards) and use the calculated acute-chronic workload ratio (ACWR) to obtain ground truth labels.

ACWR is a ratio of acute training load (total training impulse, or TRIMP, over the past 7 days) divided by chronic training load (28-day average of acute load). An ACWR equal to 1.0 means that the user has exercised at the same intensity continuously over the past week compared to the month, below 1.0 means that they are trending downward, and above 1.0 means they are trending upwards. Precise ACWR calculation involves multiple mathematical operations, so we assess the model's ability to understand the trend from monthly TRIMP values without explicit calculation.

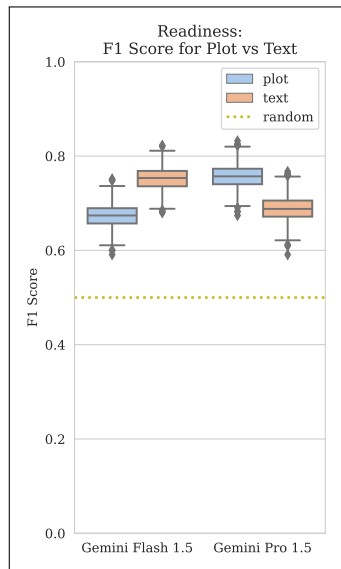

Figure 5: Results of readiness task for Gemini models only (as the dataset cannot be sent to other models), demonstrating approximate parity between the text and plot approaches.

We use training load data from 350 fitness case studies (Cosentino et al., 2024) and present it as tables or TRIMP bar plots (results shown in Figure 5). Each case study contains data from 30 consecutive days. We use a simplified version of the textual prompt and visualization from Cosentino et al. (2024) and do not split TRIMP in different heart rate zones. We tested Gemini 1.5 Pro and Gemini 1.5 Flash for both plot and text approaches as zero-shot tasks. Since this task involved analyzing just 30 data points we did not expect the plot prompt to excel here. Interestingly, models of different sizes showed opposite trends: Gemini 1.5 Pro had a slight increase in performance when using plots, while Gemini 1.5 Flash had a slight increase in performance when using text, though the magnitudes of the gains were small.

### 4.3 ABLATIONS

We considered a variety of text and plot ablations to confirm if there were any large gains. All ablations were performed on Gemini Pro 1.5. We used the function identification task to test for any performance differences; details, results and visualizations are reported in Section A.3.

### 4.4 COST

Using plots for time-series data can often be more cost-efficient and token-efficient. Token efficiency matters when your context is large and the context-window is limited; for example we needed to downsample our raw signals to fit them into the 128k context window for GPT4o(-mini), particularly with the large few-shot experiments (Section 4.2). For example, when using the Gemini API (Google, 2024), images account for 258 tokens if both dimensions are less than 384 x 384 pixels, after which 4 additional crops are added for a total of 1290 tokens. Text tasks can easily be 10x larger (e.g. 10-shot activity recognition Section 4.2) requiring more than the entire 128k context available. Depending on the task it may be possible to reduce the number of text tokens by further sub-sampling of the data, but this may result in reduced task performance. The optimal sampling rate may also be task- and dataset-specific.

Plot experiments also end up being cheaper. As an example, for our most expensive experiment on few-shot activity recognition, we can estimate the input token cost of our 5-shot experiment for both plot and text on GPT4o (OpenAI, 2024). The text version of this task required nearly 128k text tokens (costing $0.32 per 5-shot question); by contrast, the plot version required 50 images (costing $0.032), a 10x difference in overall costs for input tokens. In addition to being cheaper, the plotting approach also scales better for longer time-series, as the number of tokens required for the textual

approach grows with the number of data points while a plot of the same size will generate the same number of tokens independent of the number of data points being plotted.

# 5    CONCLUSIONS AND FUTURE WORK

The key finding from our rigorous empirical evaluation is that engaging the vision encoder of a multimodal foundation model through the use of plot representations leads to significant performance and efficiency gains on time-series understanding tasks, compared with relying on the text encoder. By processing data visually instead of textually, these models can better capture temporal patterns and relationships. We established our results on synthetic data with well-controlled characteristics and reasoning types, and also showed that this approach holds on real, noisy and complex tasks related to making sense of consumer health signals. This is analogous to the gains that humans benefit from when visualising data (Card et al., 1999; Yalçin et al., 2016), though we do not claim that the mechanisms by which visual data are processed by the models we study here are the same as those with which humans process visual stimuli.

The method presented here is powerful in its simplicity and generalizability and relies on the native capabilities of multimodal models requiring no additional training – we believe that it is particularly useful when the following conditions are met:

- You want to use an off-the-shelf multimodal model to interpret your time-series data, as might be the case in user-facing applications that rely on general models to understand a broad range of potential user inputs including natural language.
- Your use-case is not restricted to a specific task or modality, and generalizability across tasks is more important than accuracy on a single task. We showed that plots act as a generalizable time-series encoder across many tasks, even though they may not be better than a task-specific encoder trained for one task. Training task-specific encoders for multimodal models can be limited by availability of paired training data, compute and expertise.
- You don't want to downsample your data. In many cases the textual representation of real time-series outstrips the maximum context length, and so the plot-based approach is the only way to present the data without downsampling.

Our focus in this work is specific to time-series understanding (i.e., reasoning about known data). Forecasting is also important to time-series analysis and in the future we suspect that leveraging the vision components of multimodal models might yield positive results in this area too.

In this work, all plots were generated by human-written code in order to avoid any bias. As such we do not rigorously study what the optimal plotted form of a certain time-series might be for visual understanding; this is likely to be a function of the exact downstream task or user request, but could in theory be automated and forms the basis for future work. Looking forward, in real applications we anticipate that plotting could be part of a tool-use framework, where the model is prompted to choose how and when to plot the data, after which it uses the plot representation it created.

Lastly, further work remains in the explainability context – while we demonstrate empirically that visual representations generally outperform textual representations of time-series data, we have not probed why mechanistically this is so.

# 6    REPRODUCIBILITY STATEMENT

Our evaluations are run on publicly available models that have publicly available APIs. The exact model versions used are detailed in Section 3.2.

Our structured prompting methods are detailed in 3.1 and we include the actual prompts and target dataclasses used in Supplementary Section A.4.

All the data for the synthetic tasks is by definition synthesized and the detailed task descriptions in Supplementary Section A.1.1 provide the necessary details to recreate these synthetic datasets.

The IMU Fall Detection Dataset (IMUFD, Aziz et al. (2017)) and Heterogeneity Human Activity Recognition dataset (HHAR, Stisen et al. (2015)) used respectively for the fall detection and ac-

tivity recognition tasks are both publicly available. Pre- and post-processing steps are detailed in Supplementary Section A.1.2.

The dataset for the Readiness task is not currently publicly available. However the task details in Section 4.2 would enable reproduction of the results with access to a comparable dataset.

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

# A  SUPPLEMENTARY INFORMATION

## A.1  DETAILED TASK DESCRIPTIONS

In this section we provide further details of each task implementation to assist with reproducibility of results. We also provide Python code snippets for synthetic data generation and plotting.

### A.1.1  SYNTHETIC TASKS

**Functional form identification**

- We generate $(x, y)$ series of linear, quadratic, cubic, exponential and periodic functions with variable number of points and noise (injected into the function domains) over the range $x \in [-10, 10]$.

- We perform five repeats across different numbers of points (50, 500, 1000 and 2500) and noise levels (0.0, 0.5, 1.0, 2.0 and 5.0), giving 500 samples per model across number of points, noise level, function type and random replica dimensions.

- These results are then passed either as a stringified series of $x$ and $y$ vectors ("text" task), or as a matplotlib figure (the "plot" task) to the model.

- This task only requires that the model is able to understand and label the global longitudinal trend of the function, without overly needing to reason about magnitudes.

```python
def generate(rng: np.random.Generator, func_type: str,
    x_range: tuple[int, int], num_points: int,
    noise_level: float):
  x = np.linspace(x_range[0], x_range[1], num_points)
  noise = rng.normal(0, noise_level, num_points)
  if func_type == "exponential":
    y = np.exp(x + noise)
  elif func_type == "periodic":
    y = np.sin(x + noise)
  elif func_type == "quadratic":
    y = (x + noise) ** 2
  elif func_type == "linear":
    y = x + noise
  elif func_type == "cubic":
    y = (x + noise) ** 3
  else:
    raise ValueError("Invalid function type %s" % func_type)
  return x, y
```

Functional form identification - Data generation code

```python
fig = plt.figure(figsize=(6.4, 4.8)) # Rendered at 100 dpi
ax = fig.gca()
ax.scatter(xs, ys)
ax.set_title("Data showing a trend to be identified.")
ax.set_xlabel("x")
ax.set_ylabel("y")
ax.grid(True)
```

Functional form identification - Matplotlib plotting code

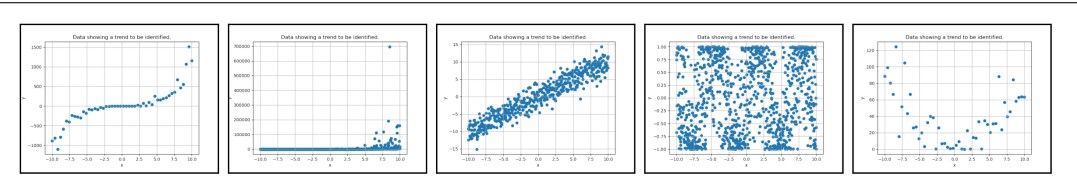

Supplementary Figure S1: Plots used for functional form identification task examples chosen at random representing at various noise levels (from left to right): cubic, exponential, linear, periodic, quadratic.

**Correlation of two lines**

- We generate $(\boldsymbol{x}, \boldsymbol{y_1})$ and $(\boldsymbol{x}, \boldsymbol{y_2})$ series that represent linear functions $y = m \cdot x$ with variable pairs of slopes $(m_1, m_2) \in \{(1, 2), (-1, 1), (5, -1), (-2, -5), (-3, 2), (2, 3)\}$.

- We perform trials across different numbers of points (50, 500, 1000 and 2500) and noise levels (0, 0.25, 0.5, 1, 1.5, 2.0, 3.0 and 5.0) over the range $\boldsymbol{x} \in [-10, 10]$, with 192 samples per model across number of points, noise level and random replica dimensions.

- These results are then passed either as a stringified series of $\boldsymbol{x}$, $\boldsymbol{y_1}$ and $\boldsymbol{y_2}$ vectors ("text" task), or as a matplotlib figure (the "plot" task) to the model. The model is then asked to classify whether the two lines $\boldsymbol{y_1}(\boldsymbol{x})$ and $\boldsymbol{y_2}(\boldsymbol{x})$ are positively or negatively correlated.

- This task requires first understanding two global trends, and then comparing them with each other.

```python
def generate(rng: np.random.Generator, x_range: tuple[int, int],
    slope_coeffs: tuple[int, int],  num_points: int,
    noise_level: float):
 x = np.linspace(x_range[0], x_range[1], num_points)
 y1_noise = rng.normal(0, noise_level, num_points)
 y2_noise = rng.normal(0, noise_level, num_points)
 y1 = slope_coeffs[0] * (x + y1_noise)
 y2 = slope_coeffs[1] * (x + y2_noise)
 return x, y1, y2
```

Correlation of two lines - Data generation code

```python
fig = plt.figure(figsize=(4, 4)) # Rendered at 96 dpi
ax = fig.gca()
ax.set_title("Synthetic Function Comparison")
ax.plot(xs, y1s, color="red", linewidth=3, label="y1")
ax.plot(xs, y2s, color="blue", linewidth=3, label="y2")
ax.legend(loc="lower right")
ax.set_xlabel("x")
ax.set_ylabel("y")
ax.grid(True)
```

Correlation of two lines - Matplotlib plotting code

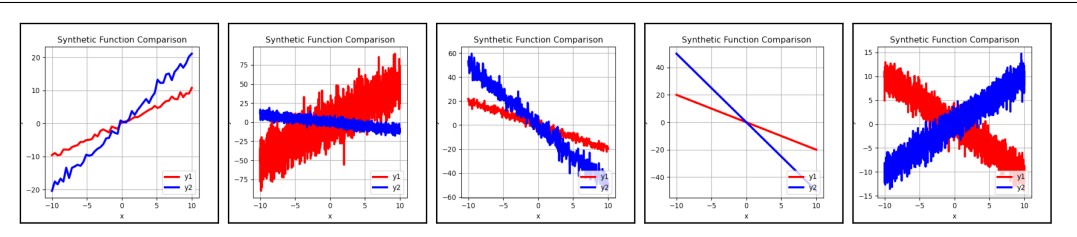

Supplementary Figure S2: Plots used for correlation of two lines task examples chosen at random representing positive and negative correlations at various noise levels.

**2D cluster counting**

- We generate a series of points corresponding to $N$ distinct clusters, parameterised by the standard deviation from the cluster center which also controls the difficulty of the task.

- The cluster centers are chosen randomly, and we enforce a minimum distance between clusters.

- We perform five repeats across different levels of standard deviation (0.025, 0.05 and 0.075), different levels of number of points per clusters (5, 50 and 100) and with the number of clusters from 1 to 9, giving 405 samples per model across standard deviation, number of clusters, number of points per clusters and random replica dimensions.

- Extending the correlation task, this task now requires that the model is able to simultaneously identify and keep separate track of $N$ different patterns.

```python
def _generate_centers(num, rng, radius = 1.0, margin = 0.1):
  def _has_close_centers(center_coordinates, distance_threshold):
    distances = scipy.spatial.distance.cdist(
        center_coordinates, center_coordinates, "euclidean")
    mask = np.triu(np.ones_like(distances), k=1).astype(bool)
    return np.any(distances[mask] < distance_threshold)
  for _ in range(10000):
    coordinates = [
        (radius * x, radius * y)
        for x, y in zip(
            rng.uniform(-radius+margin, radius-margin, num),
            rng.uniform(-radius+margin, radius-margin, num),
        )
    ]
    if not _has_close_centers(coordinates, radius * 0.3):
      return coordinates
  raise ValueError("Could not find well separated centers")

def generate(rng: np.random.Generator, seed: int,
    cluster_points: int, cluster_count: int, cluster_std: float):
  generated_samples, _, _ = sklearn.datasets.make_blobs(
      n_samples=cluster_points * cluster_count,
      centers=_generate_centers(cluster_count, rng),
      cluster_std=cluster_std,
      random_state=seed,
      return_centers=True,
      center_box=(-1, 1),
  )
  return generated_samples[:, 0], generated_samples[:, 1]
```

2D cluster counting - Data generation code

```
fig = plt.figure(figsize=(8, 8)) # Rendered at 96 dpi
ax = fig.gca()
ax.scatter(xs, ys, c="black", s=50)
ax.set_title("Synthetic Clustered Data")
ax.set_xlabel("x")
ax.set_ylabel("y")
ax.set_xlim(-1, 1)
ax.set_ylim(-1, 1)
ax.grid(True)
```

2D cluster counting - Matplotlib plotting code

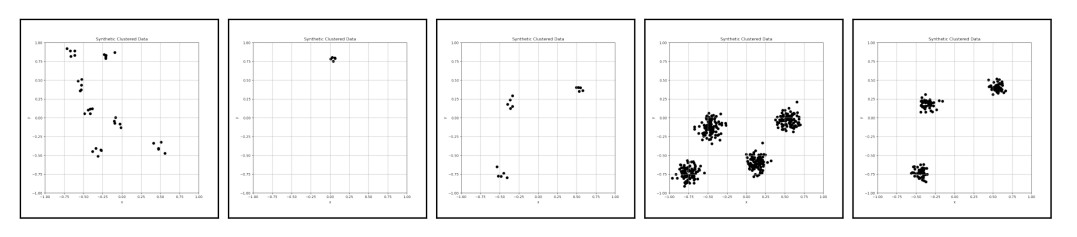

Supplementary Figure S3: Plots used for 2D cluster counting task examples chosen at random representing varying cluster parameterizations.

**Derivative identification**

- We generate $(x, y)$ series of linear, quadratic, cubic, exponential and periodic functions and their derivatives $y'(x)$ over the range $x \in [-10, 10]$.

- We perform five repeats across different numbers of points (50, 500, 1000 and 2000) and noise levels (0.0, 0.5, 1.0, 2.0 and 5.0) giving 500 samples per model across number of points, noise level, function type and random replica dimensions.

- There are four multiple choices per sample, and each choice is the derivative series $y'(x)$ of a random selection of function types, with the same noise level and number of points as the function in question.

- These results are then passed either as a stringified series of $x$ and $y$ vectors ("text" task), or as a matplotlib figure (the "plot" task) to the model.

- This task represents a multi-step extension of the function identification task: here we require the model first to understand a function, next reason about what the functional trend implies about the characteristics of its derivative, and then finally identify those characteristics within the set of multiple choices. Beyond simply introducing a multi-reasoning requirement, we also focus on derivative understanding as rates of change are key components of time-series analysis and understanding. Note that because the choices are different functional classes, the model can achieve good accuracy without reasoning about the functional magnitudes.

```
def generate(rng: np.random.Generator, func_type: str,
    x_range: tuple[int, int], num_points: int,
    noise_level: float):
  x = np.linspace(x_range[0], x_range[1], num_points)
  noise = rng.normal(0, noise_level, num_points)
  if func_type == "exponential":
    y = np.exp(x + noise)
    dy = np.exp(x + noise)
  elif func_type == "periodic":
    y = np.sin(x + noise)
```

```
    dy = np.cos(x + noise)
  elif func_type == "quadratic":
    y = (x + noise) ** 2
    dy = 2.0 * (x + noise)
  elif func_type == "linear":
    y = x + noise
    dy = np.ones(len(x)) + noise
  elif func_type == "cubic":
    y = (x + noise) ** 3
    dy = 3.0 * (x + noise) ** 2
  else:
    raise ValueError("Invalid function type %s" % func_type)
  return x, y, dy
```

Derivative identification - Data generation code

```
fig = plt.figure(figsize=(6, 4)) # Rendered at 100 dpi
ax = fig.gca()
ax.scatter(x, y)
ax.set_title(
  f"Potential derivative choice {choice_idx}" if is_derivative
  else "Function whose derivative is to be identified.")
ax.set_xlabel("x")
ax.set_ylabel("dy" if is_derivative else "y")
ax.grid(True)
```

Derivative identification - Matplotlib plotting code

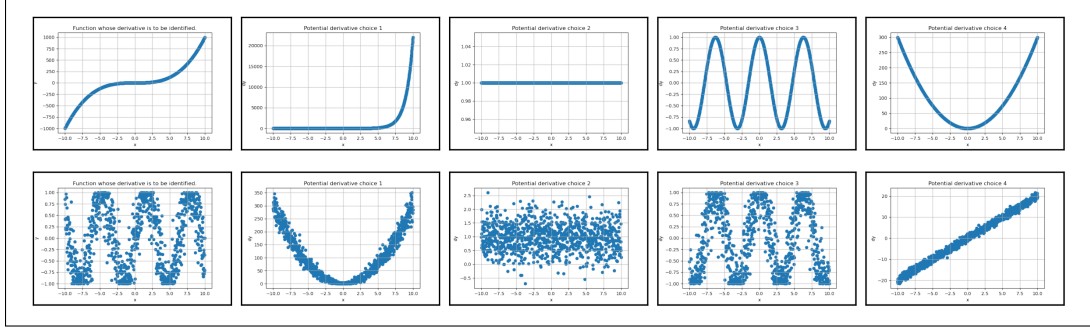

Supplementary Figure S4: Each row shows plots used for a randomly selected derivative identification task example. The leftmost plot in the row is the function to identify the derivative of, and the remaining plots are the four multiple choices.

**Quadratic derivative identification**

- We generate $(\boldsymbol{x}, \boldsymbol{y})$ series of quadratics (of form $y(x) = A \cdot x^2$) and their derivatives $\boldsymbol{y}'(\boldsymbol{x})$ over a range of scales $A \in \{-10, -5, -1, 1, 5, 10\}$ over the range $\boldsymbol{x} \in [-10, 10]$.

- We perform five repeats across different numbers of points (50, 500, 1000 and 2000) and noise levels (0.0, 0.5, 1.0, 2.0 and 5.0), with 600 samples per model and number of few-shots across number of points, noise level, function type and random replica dimensions.

- There are four multiple choices per sample, and each choice is a random selection of derivatives of quadratic functions with a range of scales $A \in \{-20, -15, -10, -5, -1, 1, 5, 10, 15, 20\}$, with the same noise level and number of points as the quadratic function in question.

- We create few-shot examples in the same manner as the main task, with the quadratic functions being randomly sampled from a range of scales $A \in \{-20, -15, -3, 3, 15, 20\}$ with 0 noise and 50 data points.

- These results are then passed either as a stringified series of $x$ and $y$ vectors ("text" task), or as a matplotlib figure (the "plot" task) to the model.

- This hard variant of the derivatives task introduces a new requirement for the model to reason about function magnitudes as all the possible choices are the correct functional form (i.e., linear).

```python
def generate(rng: np.random.Generator, quadratic_scale: float,
    x_range: tuple[int, int], num_points: int,
    noise_level: float):
  noise = rng.normal(0, noise_level, num_points)
  x = np.linspace(x_range[0], x_range[1], num_points)
  y = quadratic_scale * (x + noise) ** 2
  dy = 2.0 * quadratic_scale * (x + noise)
  return x, y, dy
```

Quadratic derivative identification - Data generation code

```python
fig = plt.figure(figsize=(6, 4)) # Rendered at 100 dpi
ax = fig.gca()
ax.scatter(x, y)
ax.set_title(
  f"Potential derivative choice {choice_idx}" if is_derivative
  else "Quadratic function whose derivative is to be identified.")
ax.set_xlabel("x")
ax.set_ylabel("dy" if is_derivative else "y")
ax.grid(True)
```

Quadratic derivative identification - Matplotlib plotting code

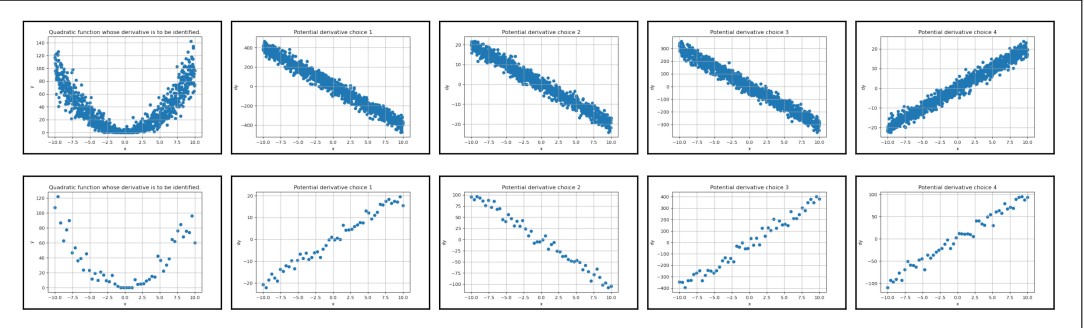

Supplementary Figure S5: Each row shows plots used for a randomly selected quadratic derivative identification task example. The leftmost plot in the row is the function to identify the derivative of, and the remaining plots are the four multiple choices.

### A.1.2 REAL-WORLD TASKS

**Fall detection**
The task is hard to define as zero-shot, since there isn't a natural way of explaining the plots and text, so we frame this as a few-shot task. We test few-shot examples with 1, 3, 5 and 10 examples, with 480 samples for each body location per model and number of few-shots. Model API errors were ignored as long as at least one body location succeeded for that example.

*Pre-processing and post-processing*

Due to context-window limitations for the text-only task, we apply a 1D average pool with a kernel size of 10 and stride of 10 to the data. This allows us to use up to 10 few-shot examples in the text-only tasks and still fit into the GPT4o and GPT4o-mini 128k context window.

The IMUFD dataset provides multiple IMUs from 7 body locations. We consider a subset of head, waist, left and right thigh. We sample IMUs from each as separate examples - when evaluating either text or plot approach, the final prediction is a majority vote from the predictions from each of these body parts.

The dataset is stratified into train (20%) and test (80%) based on participant ID. Few-shot examples are chosen from the "train" set while eval data is chosen from the "test" set. This ensures the model never sees any data from the same participant.

*Plotting*

```python
fig = plt.figure(figsize=(6, 4)) # Rendered at 100 dpi
ax = fig.gca()
for i in range(6):
  ax.plot(example[i])
```

Fall detection - Matplotlib plotting code

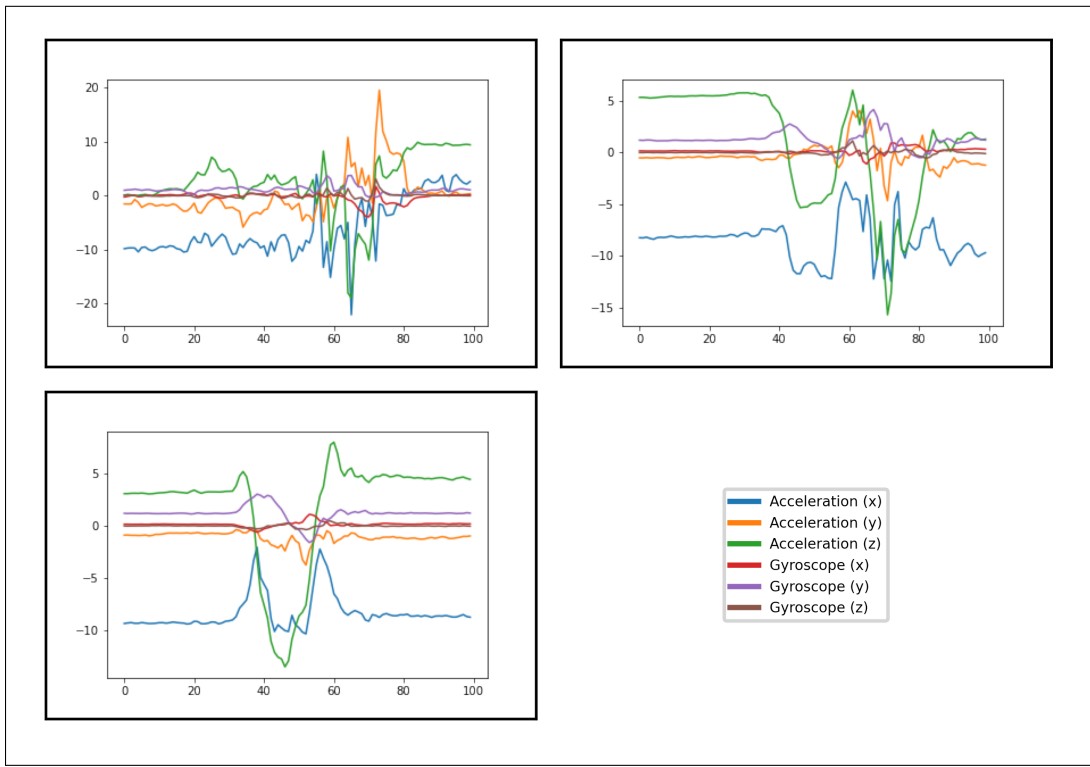

Supplementary Figure S6: Illustrative plots of example entries from the IMU Fall Detection Dataset (IMUFD, Aziz et al. (2017)) that were correctly categorized by Gemini Pro (10-shot). Top-Left: Fall. Top-Right: Near Fall. Bottom-Left: active daily living (ADL). Examples were collected with an IMU sensor at the waist and each of the three acceleration and gyroscope axes were plotted as a different series on the same plot. The legend was not provided to the model, but is shown here to aid visualisation.

*Comparison with task-specific model*

In Table S1 we compare the results from most performant foundation models studied here (GPT4o and Gemini Pro 1.5) with a task-specific model reported in (Aziz et al., 2017). While we don't

achieve the same sensitivity and specificity as expected for a general model, our plot results are of the same order of magnitude.

| Model | Sensitivity | Specificity |
|---|---|---|
| GPT4o, plot, 10-shot | 0.92 | 0.81 |
| GPT4o, text, 10-shot | 0.70 | 0.49 |
| Gemini Pro 1.5, plot, 10-shot | 0.84 | 0.95 |
| Gemini Pro 1.5, text, 10-shot | 0.61 | 0.70 |
| Task-specific SVM | 0.96 | 0.96 |

Supplementary Table S1: Comparison of most performant foundation models with task-specific fall detection support vector machine (SVM) as reported in (Aziz et al., 2017).

**Activity recognition**

*Pre-processing*

As with the fall detection data, we apply a 1D average pool over the raw data to limit the number of text tokens. As the raw IMU sample rates varied between 70-200Hz the kernel size, and matching stride, was chosen to target a downsampled frequency of 10Hz for each example. We select few-shot examples using leave-one-out cross-validation at the dataset user level to maximise the number of examples used for validation while ensuring we exclude any few-shot examples from the same user, even those from different device types.

The raw HHAR dataset consists of examples with varying durations and longer examples typically contain multi-second gaps with no samples from the IMU. We chunk each raw example by splitting on any gaps longer than 2 seconds and take a central 15 second crop of the longest chunk to create the examples used in this study. Any examples with mismatching sample rates for the accelerometer and gyroscope are filtered out. The raw labels "stairsup" and "stairsdown" are coalesced into a single "stairs" class.

*Plotting*

For the activity recognition task, we plot accelerometer and gyroscope signals separately – we find empirically that plotting the signals separately has a marginally beneficial effect on performance over plotting together (see Supplementary Figure S8).

```python
figs = []
for sensor_label, idxs in [
    ("Accelerometer", [1, 2, 3]),
    ("Gyroscope", [4, 5, 6])]:
  fig = plt.figure(figsize=(4, 4)) # Rendered at 90 dpi
  ax = fig.gca()
  ax.set_title(sensor_label)
  for axis, idx in zip(["x", "y", "z"], idxs):
    ax.plot(example[0], example[idx], label=axis)
  ax.legend()
  ax.set_xlabel("Time (s)")
  ax.set_xlim(0, 15)
  figs.append(fig)
```

Activity recognition - Matplotlib plotting code

1080
1081
1082
1083
1084
1085
1086
1087
1088
1089
1090
1091
1092
1093
1094
1095
1096
1097
1098
1099
1100
1101
1102
1103
1104
1105
1106
1107
1108
1109
1110
1111
1112
1113
1114
1115
1116
1117
1118
1119
1120
1121
1122
1123
1124
1125
1126
1127
1128
1129
1130
1131
1132
1133

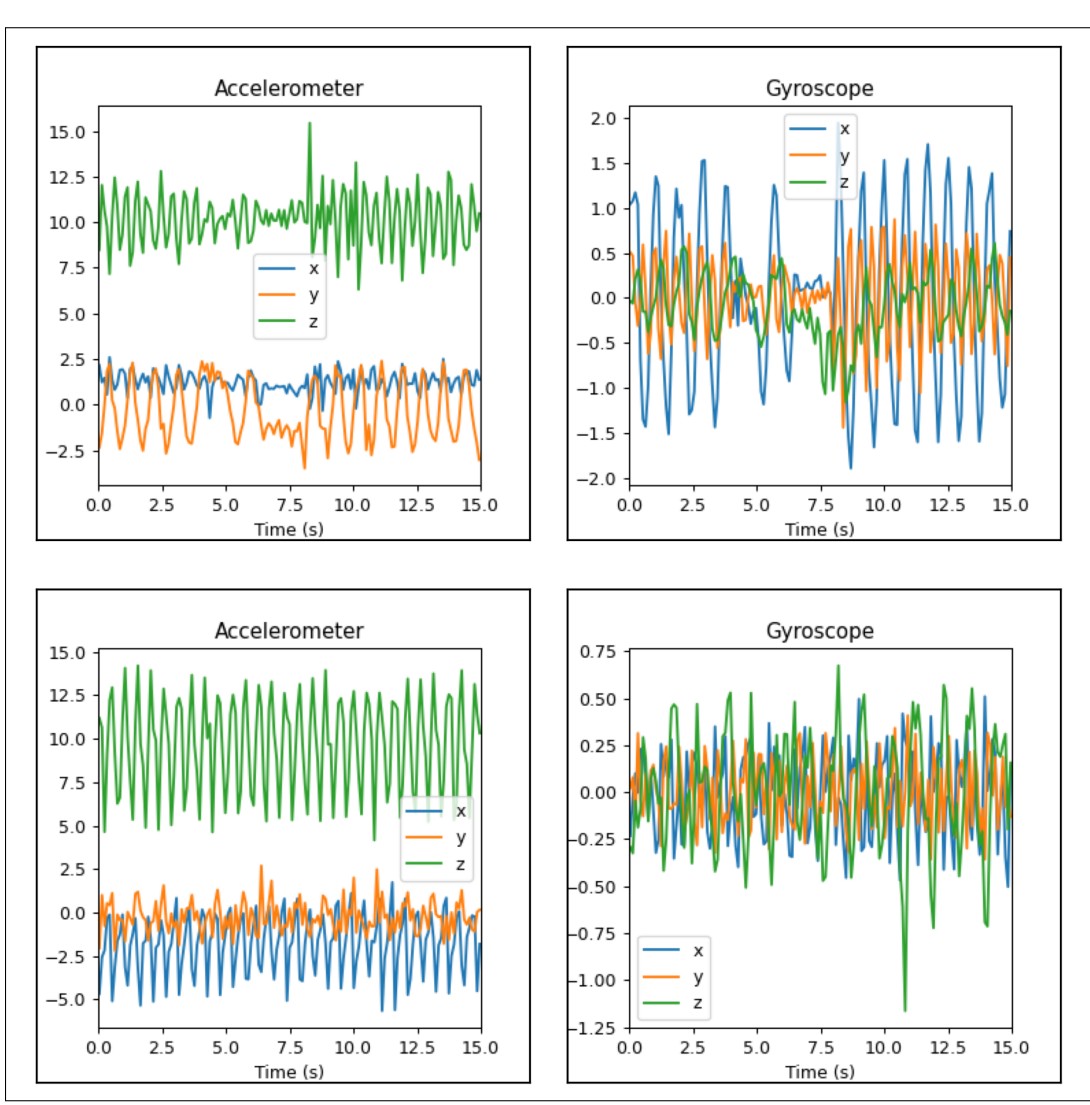

Supplementary Figure S7: Illustrative plots of example entries from the Heterogeneity Human Activity Recognition dataset (HHAR, Stisen et al. (2015)) that were correctly categorized by Gemini Pro (10-shot). Top: Bike. Bottom: Walk. These examples were collected from a Nexus4 device. The signals for each sensor type are plotted separately.

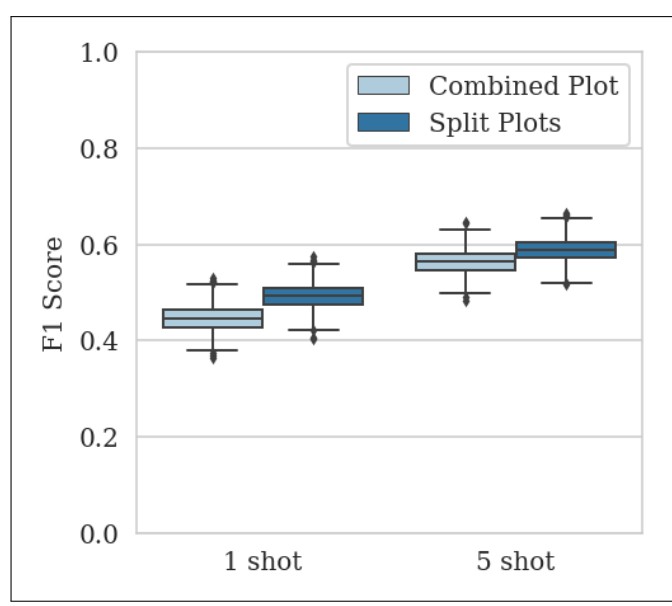

Supplementary Figure S8: Performance comparison between combining all 6 IMU axes in a single plot vs splitting the accelerometer and gyroscope into distinct plots for the Activity Recognition task. Results shown for 1-shot and 5-shot on Gemini Pro indicate a marginal performance increase when the data is split into 2 plots, which is reduced when moving from 1-shot to 5-shot.

## A.2 FURTHER RESULTS

We first present the relative differences in task performance between plot and text approaches in Supplementary Table S2.

| Task (Metric) | Shots | GPT4o-mini | Gemini Flash 1.5 | GPT4o | Gemini Pro 1.5 |
|---|---|---|---|---|---|
| Functional form id. (Accuracy) | 0 | 70.8 | 82.4 | 122.7 | 11.1 |
| Correlation of two lines (Accuracy) | 0 | 66.7 | 42.9 | 20.0 | 50.0 |
| 2D cluster counting (MAE) | 0 | 55.8 | 65.5 | 86.4 | 85.0 |
| Derivative id. (Accuracy) | 0 | 40.0 | 31.6 | -5.3 | -0.0 |
| Quadratic derivative id. (Accuracy) | 0 | 100.0 | 85.7 | -37.5 | 47.4 |
| | 1 | 37.5 | 107.1 | 23.8 | 113.3 |
| | 2 | 92.9 | 100.0 | 8.0 | 128.6 |
| | 3 | 108.3 | 112.5 | 9.1 | 172.7 |
| Fall detection ($F_1$) | 1 | 11.6 | 36.5 | 114.4 | 27.3 |
| | 10 | 120.6 | 49.1 | 153.0 | 92.4 |
| Activity detection ($F_1$) | 1 | 36.0 | 36.5 | 6.3 | 69.3 |
| | 5 | 48.0 | 77.2 | 39.7 | 64.6 |
| Readiness ($F_1$) | 0 | – | -10.8 | – | 10.0 |

Supplementary Table S2: Percent differences in median plot performances relative to median text performances.

Next, we present results of the synthetic task performances as functions of the various dataset parameters that control the difficulty of the task example. The variable model responses over different combinations of the dataset parameters described in this section create the distributions shown in Figure 1. Depending on the task, these dataset parameters include:

- **Number of points**: the number of points per series. For all tasks except for 2D cluster counting, this means the number of function samples between $x = -10$ and $x = 10$. For the cluster counting task, this is simply the number of points per cluster.

- **Noise level**: the amount of noise injected into the functions, i.e. $y = y(x + noise\_level)$. This parameter is relevant for all synthetic tasks except the 2D cluster counting.

- **Standard deviation**: for the 2D cluster counting task, this controls the tightness of each cluster.

We also show the performance of each task over the set of possible correct answers specific to that task.

### A.2.1 FUNCTIONAL FORM IDENTIFICATION

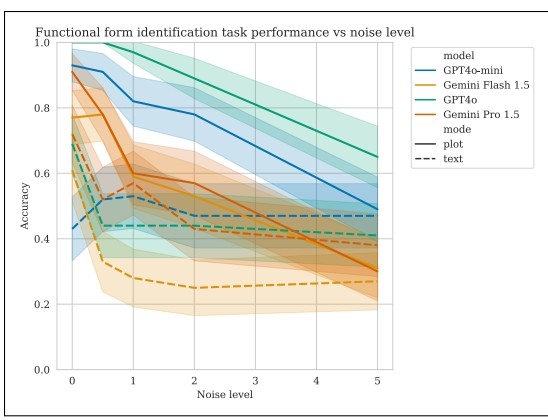

(a) Results as a function of number of points

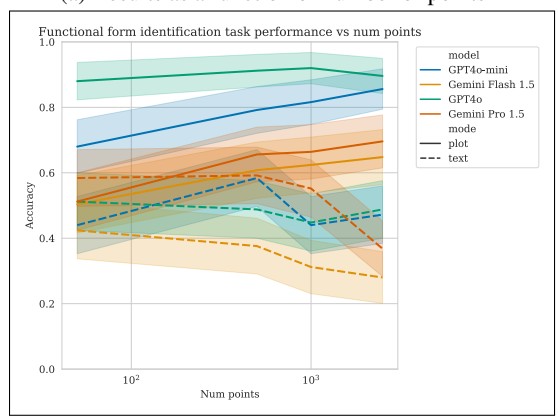

(b) Results as a function of noise level

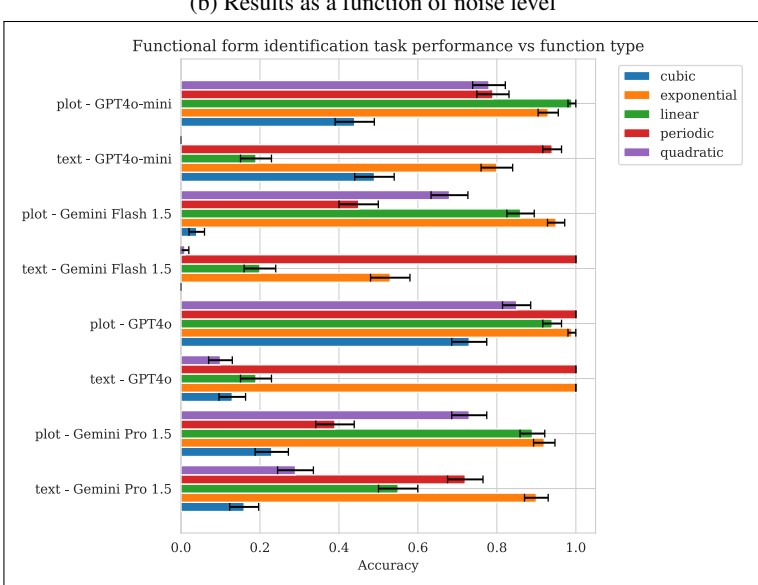

(c) Results as a function of true function type

Supplementary Figure S9: Functional form identification performance over various dataset parameters

### A.2.2 CORRELATION OF TWO LINES

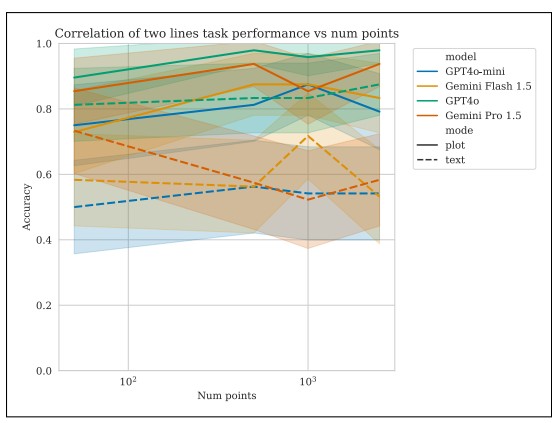

(a) Results as a function of number of points

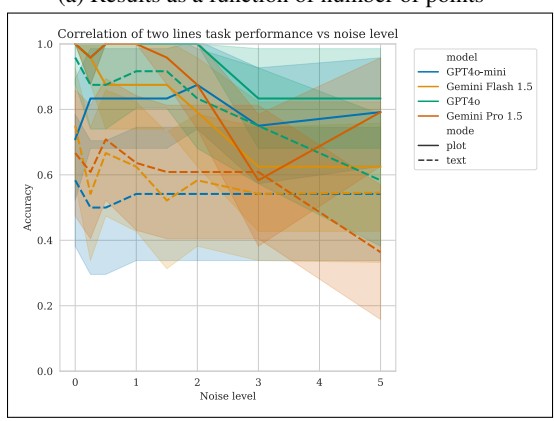

(b) Results as a function of noise level

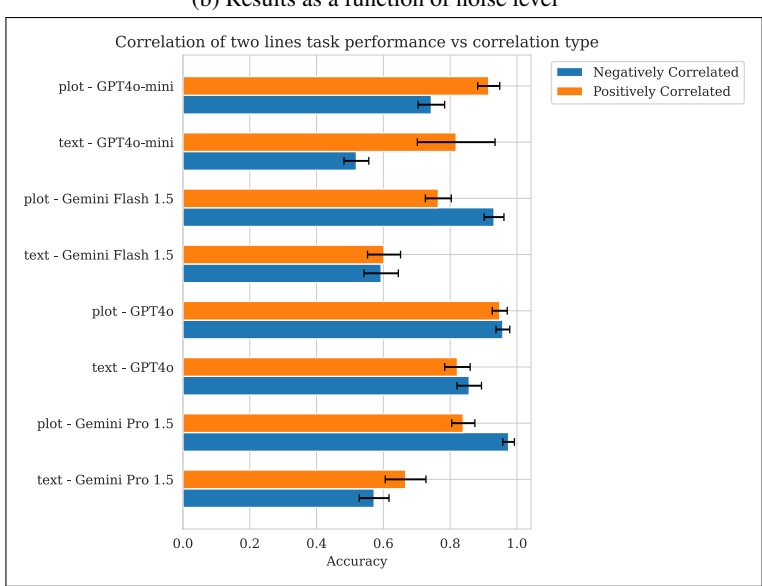

(c) Results as a function of true correlation type

Supplementary Figure S10: Correlation of two lines performance over various dataset parameters

### A.2.3 2D CLUSTER COUNTING

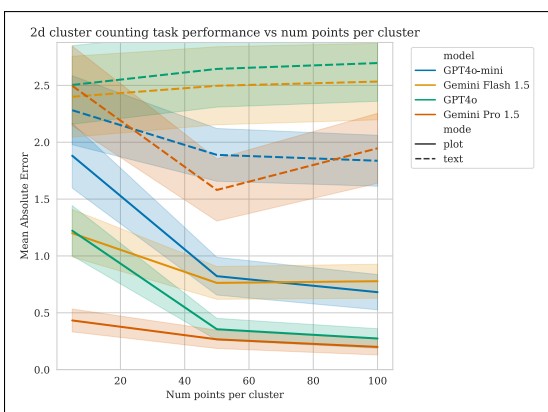

(a) Results as a function of number of points

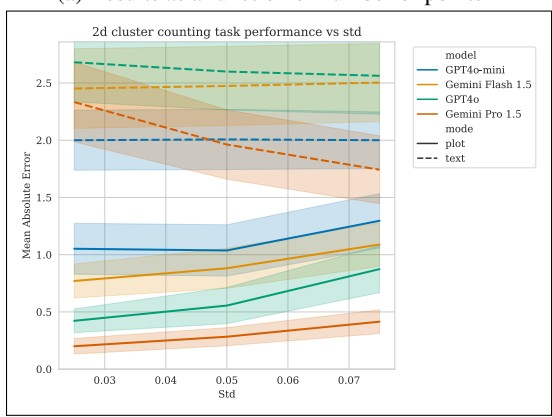

(b) Results as a function of cluster standard deviation

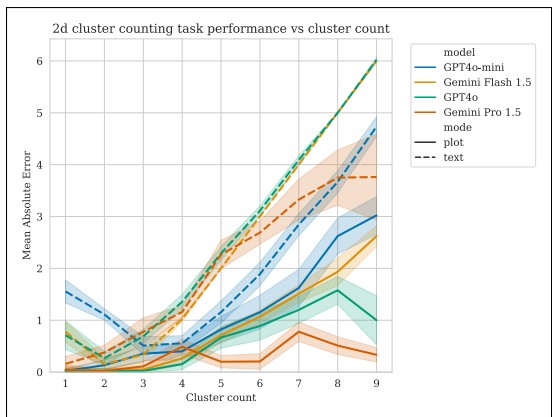

(c) Results as a function of true number of clusters

Supplementary Figure S11: 2D cluster counting performance over various dataset parameters (lower is better)

### A.2.4 DERIVATIVE IDENTIFICATION

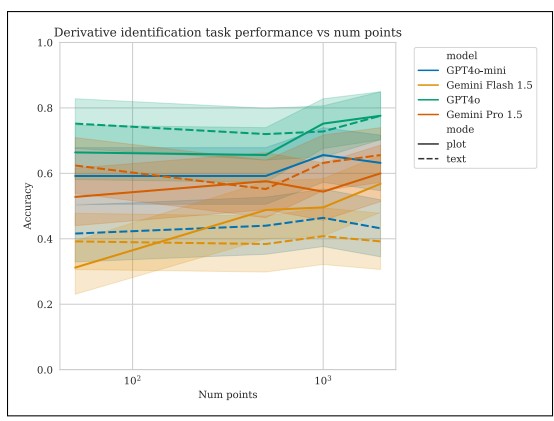

(a) Results as a function of number of points

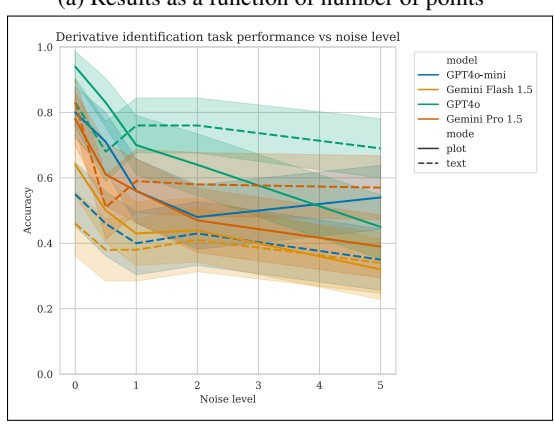

(b) Results as a function of noise level

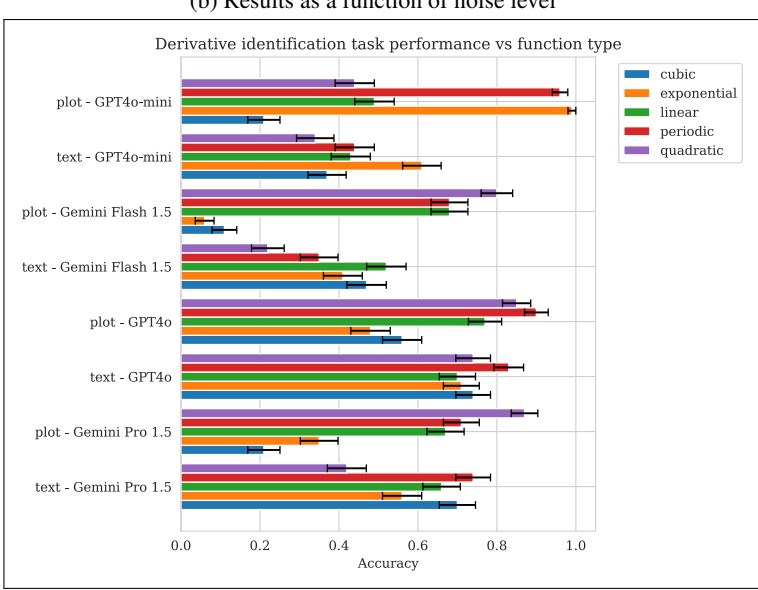

(c) Results as a function of input function class

Supplementary Figure S12: Derivative identification performance over various dataset parameters

### A.2.5 QUADRATIC DERIVATIVE IDENTIFICATION

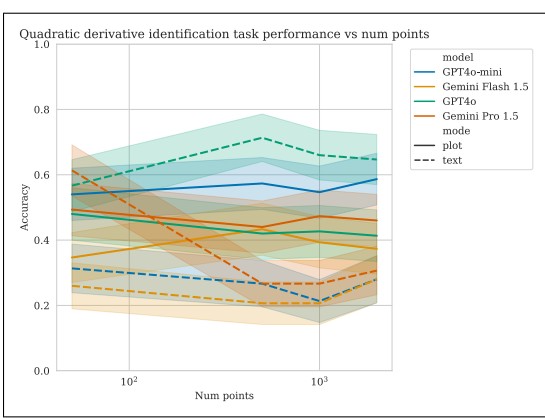

(a) Results as a function of number of points

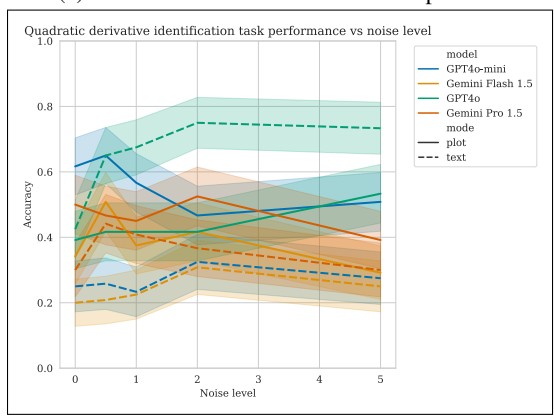

(b) Results as a function of noise level

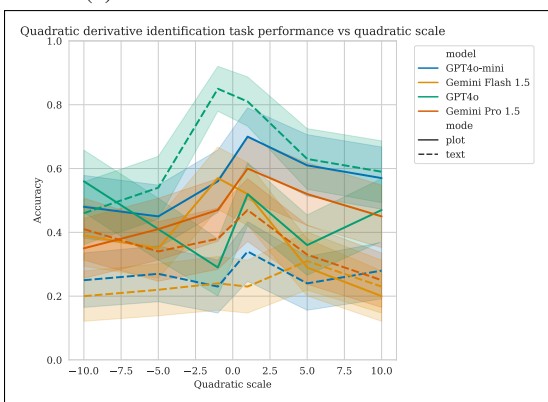

(c) Results as a function of input quadratic scale ($A$ in $y = A \cdot x^2$)

Supplementary Figure S13: Zero-shot quadratic derivative identification performance over various dataset parameters

## A.3 ABLATIONS

### A.3.1 METHODOLOGY

We perform the ablations described here on the functional form identification task using Gemini Pro 1.5.

For the text-only task we tested the effect of comma versus space separation of numbers. We were also guided by the approach in LLMTime (Gruver et al., 2024) and tested the effect of fixed precision of floating point numbers (2, 4, 8, 16) and rescaling the input numbers.

For the plot tasks, we considered the following ablations, with Supplementary Figure S14 showing examples:

- Resolution (in dpi): 25, 50, 100, 200, 400
- Figure size: $(3.5, 3.5), (4, 3), (7, 7), (8, 6), (12, 12)$
- Plot style: Default, classic, ggplot, seaborn-whitegrid, seaborn-darkgrid
- Different color palettes for text, background and scatter
- Marker types: circle, square, triangle, x-mark, plus
- Marker sizes: small (10), medium (50), large (100)
- Plot components: all (title, axis labels, spines, ticks, grid, axes), minimal (grid and axes only) or none
- Temperature: 0.0, 0.1, 0.3, 0.55, 1.0

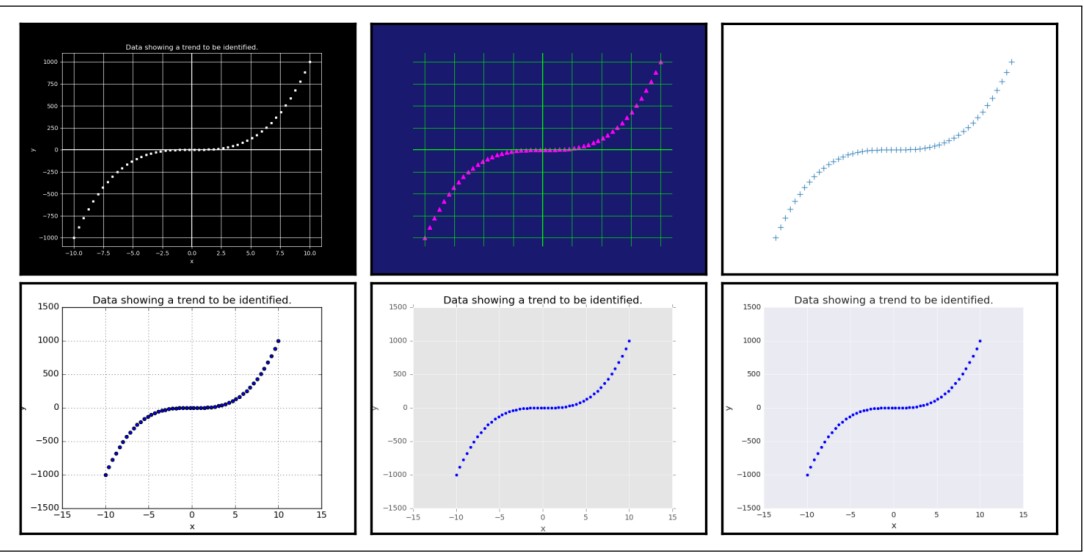

Supplementary Figure S14: A sample of the different ablations of plots used for a cubic function identification task.

We also test the combined plot and text version of the task, across both possible prompt orderings (i.e., text followed by plot, and plot followed by text).

Table cells contain mean accuracies with 95% confidence interval derived from 1,000 bootstrap repeats in brackets; rows and columns refer to different combinations of ablation and dataset parameters. We note that generally speaking, nothing had a strong positive effect on the results.

### A.3.2 TEXT-BASED RESULTS

| Separator | 50 | 500 | 1000 | 2500 |
|---|---|---|---|---|
| Comma and space | 0.63 (0.59, 0.67) | 0.50 (0.46, 0.54) | 0.50 (0.46, 0.54) | 0.36 (0.32, 0.40) |
| Space | 0.62 (0.58, 0.66) | 0.54 (0.50, 0.59) | 0.48 (0.44, 0.53) | 0.36 (0.32, 0.40) |

Supplementary Table S3: Text-based functional form identification ablation results: modifying value separator for different numbers of points.

| Separator | 0.0 | 0.5 | 1.0 | 2.0 | 5.0 |
|---|---|---|---|---|---|
| Comma and space | 0.71 (0.66, 0.75) | 0.49 (0.44, 0.54) | 0.44 (0.39, 0.49) | 0.46 (0.41, 0.51) | 0.39 (0.34, 0.44) |
| Space | 0.73 (0.69, 0.78) | 0.49 (0.44, 0.54) | 0.46 (0.41, 0.51) | 0.42 (0.37, 0.48) | 0.40 (0.34, 0.44) |

Supplementary Table S4: Text-based functional form identification ablation results: modifying value separator for different noise levels.

| Separator | cubic | exponential | linear | periodic | quadratic |
|---|---|---|---|---|---|
| Comma and space | 0.12 (0.08, 0.14) | 0.91 (0.88, 0.94) | 0.48 (0.43, 0.53) | 0.73 (0.69, 0.78) | 0.26 (0.21, 0.30) |
| Space | 0.10 (0.07, 0.12) | 0.83 (0.80, 0.87) | 0.45 (0.40, 0.50) | 0.87 (0.84, 0.91) | 0.26 (0.22, 0.30) |

Supplementary Table S5: Text-based functional form identification ablation results: modifying value separator for different function classes.

| Fixed precision | 50 | 500 | 1000 | 2500 |
|---|---|---|---|---|
| 2 | 0.70 (0.64, 0.75) | 0.53 (0.47, 0.59) | 0.51 (0.45, 0.57) | 0.31 (0.25, 0.37) |
| 4 | 0.62 (0.55, 0.68) | 0.53 (0.47, 0.59) | 0.50 (0.44, 0.56) | 0.40 (0.34, 0.46) |
| 8 | 0.60 (0.54, 0.66) | 0.51 (0.45, 0.57) | 0.52 (0.46, 0.58) | 0.36 (0.30, 0.42) |
| 16 | 0.59 (0.53, 0.64) | 0.52 (0.46, 0.59) | 0.44 (0.38, 0.51) | 0.36 (0.31, 0.43) |

Supplementary Table S6: Text-based functional form identification ablation results: modifying floating point fixed precision for different numbers of points.

| Fixed precision | 0.0 | 0.5 | 1.0 | 2.0 | 5.0 |
|---|---|---|---|---|---|
| 2 | 0.77 (0.71, 0.82) | 0.51 (0.43, 0.57) | 0.45 (0.38, 0.52) | 0.43 (0.36, 0.51) | 0.40 (0.33, 0.47) |
| 4 | 0.75 (0.68, 0.81) | 0.46 (0.40, 0.53) | 0.46 (0.40, 0.53) | 0.46 (0.39, 0.53) | 0.43 (0.36, 0.50) |
| 8 | 0.69 (0.62, 0.75) | 0.45 (0.38, 0.52) | 0.48 (0.41, 0.55) | 0.47 (0.40, 0.53) | 0.41 (0.34, 0.48) |
| 16 | 0.68 (0.61, 0.74) | 0.56 (0.49, 0.63) | 0.42 (0.35, 0.49) | 0.41 (0.34, 0.47) | 0.33 (0.27, 0.40) |

Supplementary Table S7: Text-based functional form identification ablation results: modifying floating point fixed precision for different noise levels.

| Fixed precision | cubic | exponential | linear | periodic | quadratic |
|---|---|---|---|---|---|
| 2 | 0.12 (0.07, 0.16) | 0.95 (0.92, 0.98) | 0.58 (0.52, 0.65) | 0.55 (0.48, 0.62) | 0.35 (0.29, 0.42) |
| 4 | 0.10 (0.07, 0.15) | 0.95 (0.93, 0.98) | 0.41 (0.35, 0.48) | 0.81 (0.76, 0.86) | 0.27 (0.20, 0.32) |
| 8 | 0.07 (0.04, 0.12) | 0.84 (0.79, 0.89) | 0.45 (0.38, 0.51) | 0.89 (0.84, 0.93) | 0.24 (0.18, 0.30) |
| 16 | 0.12 (0.08, 0.17) | 0.74 (0.68, 0.80) | 0.40 (0.33, 0.47) | 0.96 (0.94, 0.98) | 0.17 (0.12, 0.23) |

Supplementary Table S8: Text-based functional form identification ablation results: modifying floating point fixed precision for different function classes.

| rescaling | 50 | 500 | 1000 | 2500 |
|---|---|---|---|---|
| llmtime ($\alpha = 0.5, \beta = 0.0$) | 0.61 (0.52, 0.69) | 0.31 (0.23, 0.40) | 0.32 (0.24, 0.40) | 0.40 (0.32, 0.48) |
| llmtime ($\alpha = 0.5, \beta = 0.15$) | 0.55 (0.46, 0.65) | 0.38 (0.30, 0.46) | 0.35 (0.27, 0.43) | 0.30 (0.22, 0.38) |
| llmtime ($\alpha = 0.5, \beta = 0.3$) | 0.50 (0.42, 0.59) | 0.37 (0.28, 0.46) | 0.33 (0.26, 0.41) | 0.27 (0.19, 0.35) |
| llmtime ($\alpha = 0.5, \beta = 0.5$) | 0.57 (0.48, 0.66) | 0.38 (0.29, 0.46) | 0.34 (0.26, 0.43) | 0.30 (0.22, 0.38) |
| llmtime ($\alpha = 0.7, \beta = 0.0$) | 0.57 (0.48, 0.66) | 0.32 (0.24, 0.41) | 0.29 (0.21, 0.38) | 0.38 (0.30, 0.47) |
| llmtime ($\alpha = 0.7, \beta = 0.15$) | 0.54 (0.46, 0.62) | 0.35 (0.27, 0.44) | 0.30 (0.22, 0.38) | 0.30 (0.22, 0.38) |
| llmtime ($\alpha = 0.7, \beta = 0.3$) | 0.57 (0.48, 0.66) | 0.34 (0.26, 0.43) | 0.30 (0.22, 0.38) | 0.30 (0.22, 0.38) |
| llmtime ($\alpha = 0.7, \beta = 0.5$) | 0.54 (0.45, 0.62) | 0.40 (0.32, 0.49) | 0.32 (0.24, 0.40) | 0.28 (0.21, 0.36) |
| llmtime ($\alpha = 0.9, \beta = 0.0$) | 0.52 (0.43, 0.61) | 0.33 (0.24, 0.41) | 0.29 (0.20, 0.37) | 0.38 (0.30, 0.47) |
| llmtime ($\alpha = 0.9, \beta = 0.15$) | 0.52 (0.43, 0.61) | 0.38 (0.30, 0.46) | 0.32 (0.25, 0.40) | 0.27 (0.19, 0.35) |
| llmtime ($\alpha = 0.9, \beta = 0.3$) | 0.50 (0.41, 0.58) | 0.31 (0.23, 0.39) | 0.28 (0.21, 0.36) | 0.30 (0.23, 0.38) |
| llmtime ($\alpha = 0.9, \beta = 0.5$) | 0.53 (0.44, 0.62) | 0.32 (0.24, 0.39) | 0.32 (0.24, 0.40) | 0.31 (0.23, 0.39) |
| llmtime ($\alpha = 0.99, \beta = 0.0$) | 0.56 (0.47, 0.65) | 0.31 (0.23, 0.40) | 0.31 (0.24, 0.39) | 0.41 (0.33, 0.50) |
| llmtime ($\alpha = 0.99, \beta = 0.15$) | 0.54 (0.46, 0.62) | 0.34 (0.25, 0.42) | 0.38 (0.29, 0.46) | 0.32 (0.24, 0.40) |
| llmtime ($\alpha = 0.99, \beta = 0.3$) | 0.54 (0.46, 0.63) | 0.33 (0.24, 0.41) | 0.32 (0.24, 0.40) | 0.29 (0.22, 0.38) |
| llmtime ($\alpha = 0.99, \beta = 0.5$) | 0.51 (0.43, 0.60) | 0.34 (0.26, 0.42) | 0.33 (0.25, 0.41) | 0.30 (0.22, 0.37) |
| minmax | 0.61 (0.53, 0.70) | 0.42 (0.34, 0.51) | 0.35 (0.27, 0.43) | 0.24 (0.17, 0.32) |
| none | 0.62 (0.54, 0.70) | 0.58 (0.49, 0.66) | 0.57 (0.47, 0.66) | 0.35 (0.27, 0.43) |

Supplementary Table S9: Text-based functional form identification ablation results: rescaling values as suggested in (Gruver et al., 2024) for different numbers of points.

| rescaling | 0.0 | 0.5 | 1.0 | 2.0 | 5.0 |
|---|---|---|---|---|---|
| llmtime ($\alpha = 0.5, \beta = 0.0$) | 0.41 (0.31, 0.50) | 0.43 (0.33, 0.53) | 0.43 (0.33, 0.52) | 0.39 (0.29, 0.48) | 0.39 (0.30, 0.49) |
| llmtime ($\alpha = 0.5, \beta = 0.15$) | 0.48 (0.38, 0.58) | 0.42 (0.32, 0.51) | 0.40 (0.31, 0.50) | 0.36 (0.26, 0.45) | 0.31 (0.22, 0.40) |
| llmtime ($\alpha = 0.5, \beta = 0.3$) | 0.48 (0.38, 0.57) | 0.42 (0.33, 0.52) | 0.42 (0.32, 0.52) | 0.28 (0.19, 0.37) | 0.24 (0.16, 0.33) |
| llmtime ($\alpha = 0.5, \beta = 0.5$) | 0.49 (0.39, 0.59) | 0.44 (0.34, 0.54) | 0.42 (0.33, 0.51) | 0.36 (0.27, 0.46) | 0.27 (0.18, 0.36) |
| llmtime ($\alpha = 0.7, \beta = 0.0$) | 0.33 (0.24, 0.42) | 0.44 (0.34, 0.54) | 0.43 (0.33, 0.53) | 0.35 (0.26, 0.44) | 0.40 (0.31, 0.49) |
| llmtime ($\alpha = 0.7, \beta = 0.15$) | 0.45 (0.35, 0.54) | 0.47 (0.37, 0.56) | 0.34 (0.25, 0.44) | 0.34 (0.25, 0.43) | 0.26 (0.18, 0.34) |
| llmtime ($\alpha = 0.7, \beta = 0.3$) | 0.43 (0.34, 0.53) | 0.39 (0.30, 0.48) | 0.37 (0.27, 0.47) | 0.38 (0.29, 0.47) | 0.31 (0.22, 0.40) |
| llmtime ($\alpha = 0.7, \beta = 0.5$) | 0.49 (0.38, 0.58) | 0.47 (0.36, 0.57) | 0.41 (0.31, 0.50) | 0.30 (0.20, 0.39) | 0.25 (0.17, 0.33) |
| llmtime ($\alpha = 0.9, \beta = 0.0$) | 0.39 (0.30, 0.49) | 0.41 (0.31, 0.50) | 0.42 (0.32, 0.52) | 0.38 (0.29, 0.48) | 0.30 (0.21, 0.39) |
| llmtime ($\alpha = 0.9, \beta = 0.15$) | 0.45 (0.35, 0.55) | 0.40 (0.32, 0.49) | 0.39 (0.30, 0.49) | 0.35 (0.26, 0.44) | 0.27 (0.19, 0.36) |
| llmtime ($\alpha = 0.9, \beta = 0.3$) | 0.39 (0.29, 0.49) | 0.41 (0.31, 0.51) | 0.35 (0.26, 0.44) | 0.34 (0.25, 0.43) | 0.25 (0.17, 0.33) |
| llmtime ($\alpha = 0.9, \beta = 0.5$) | 0.44 (0.34, 0.54) | 0.45 (0.35, 0.54) | 0.44 (0.34, 0.53) | 0.33 (0.24, 0.43) | 0.19 (0.12, 0.27) |
| llmtime ($\alpha = 0.99, \beta = 0.0$) | 0.41 (0.32, 0.50) | 0.43 (0.33, 0.53) | 0.38 (0.29, 0.48) | 0.38 (0.29, 0.47) | 0.39 (0.29, 0.48) |
| llmtime ($\alpha = 0.99, \beta = 0.15$) | 0.50 (0.40, 0.60) | 0.43 (0.34, 0.53) | 0.44 (0.35, 0.53) | 0.30 (0.21, 0.40) | 0.30 (0.21, 0.40) |
| llmtime ($\alpha = 0.99, \beta = 0.3$) | 0.43 (0.34, 0.54) | 0.44 (0.35, 0.53) | 0.40 (0.31, 0.49) | 0.27 (0.19, 0.36) | 0.31 (0.22, 0.40) |
| llmtime ($\alpha = 0.99, \beta = 0.5$) | 0.47 (0.38, 0.57) | 0.45 (0.35, 0.55) | 0.40 (0.31, 0.50) | 0.27 (0.18, 0.36) | 0.25 (0.17, 0.34) |
| minmax | 0.39 (0.30, 0.49) | 0.53 (0.43, 0.62) | 0.48 (0.38, 0.58) | 0.35 (0.26, 0.44) | 0.28 (0.20, 0.36) |
| none | 0.77 (0.68, 0.85) | 0.54 (0.44, 0.64) | 0.55 (0.45, 0.64) | 0.45 (0.35, 0.55) | 0.34 (0.25, 0.44) |

Supplementary Table S10: Text-based functional form identification ablation results: rescaling values as suggested in (Gruver et al., 2024) for different noise levels.

| rescaling | cubic | exponential | linear | periodic | quadratic |
|---|---|---|---|---|---|
| llmtime ($\alpha$=0.5, $\beta$=0.0) | 0.01 (0.00, 0.03) | 0.68 (0.58, 0.76) | 0.50 (0.40, 0.60) | 0.62 (0.53, 0.72) | 0.24 (0.15, 0.33) |
| llmtime ($\alpha$=0.5, $\beta$=0.15) | 0.01 (0.00, 0.03) | 0.58 (0.49, 0.68) | 0.55 (0.45, 0.64) | 0.58 (0.49, 0.67) | 0.25 (0.17, 0.34) |
| llmtime ($\alpha$=0.5, $\beta$=0.3) | 0.01 (0.00, 0.03) | 0.59 (0.50, 0.69) | 0.42 (0.33, 0.51) | 0.57 (0.47, 0.67) | 0.25 (0.17, 0.33) |
| llmtime ($\alpha$=0.5, $\beta$=0.5) | 0.01 (0.00, 0.03) | 0.56 (0.46, 0.65) | 0.52 (0.42, 0.62) | 0.60 (0.50, 0.69) | 0.29 (0.21, 0.38) |
| llmtime ($\alpha$=0.7, $\beta$=0.0) | 0.02 (0.00, 0.05) | 0.71 (0.62, 0.79) | 0.44 (0.34, 0.54) | 0.62 (0.52, 0.72) | 0.16 (0.09, 0.24) |
| llmtime ($\alpha$=0.7, $\beta$=0.15) | 0.00 (0.00, 0.00) | 0.54 (0.44, 0.64) | 0.47 (0.38, 0.56) | 0.61 (0.51, 0.70) | 0.24 (0.16, 0.32) |
| llmtime ($\alpha$=0.7, $\beta$=0.3) | 0.00 (0.00, 0.00) | 0.62 (0.52, 0.72) | 0.46 (0.36, 0.56) | 0.57 (0.47, 0.67) | 0.23 (0.15, 0.31) |
| llmtime ($\alpha$=0.7, $\beta$=0.5) | 0.03 (0.00, 0.07) | 0.57 (0.47, 0.67) | 0.49 (0.40, 0.59) | 0.59 (0.49, 0.69) | 0.24 (0.16, 0.33) |
| llmtime ($\alpha$=0.9, $\beta$=0.0) | 0.01 (0.00, 0.03) | 0.68 (0.59, 0.77) | 0.36 (0.27, 0.45) | 0.61 (0.52, 0.70) | 0.24 (0.16, 0.32) |
| llmtime ($\alpha$=0.9, $\beta$=0.15) | 0.00 (0.00, 0.00) | 0.61 (0.52, 0.70) | 0.48 (0.39, 0.58) | 0.57 (0.47, 0.67) | 0.20 (0.12, 0.28) |
| llmtime ($\alpha$=0.9, $\beta$=0.3) | 0.02 (0.00, 0.05) | 0.48 (0.38, 0.57) | 0.40 (0.31, 0.50) | 0.65 (0.55, 0.74) | 0.19 (0.12, 0.27) |
| llmtime ($\alpha$=0.9, $\beta$=0.5) | 0.03 (0.00, 0.07) | 0.53 (0.43, 0.63) | 0.53 (0.44, 0.63) | 0.53 (0.43, 0.63) | 0.23 (0.15, 0.32) |
| llmtime ($\alpha$=0.99, $\beta$=0.0) | 0.01 (0.00, 0.03) | 0.71 (0.62, 0.80) | 0.40 (0.30, 0.50) | 0.62 (0.53, 0.71) | 0.25 (0.17, 0.34) |
| llmtime ($\alpha$=0.99, $\beta$=0.15) | 0.00 (0.00, 0.00) | 0.59 (0.48, 0.69) | 0.48 (0.38, 0.58) | 0.63 (0.54, 0.72) | 0.27 (0.18, 0.36) |
| llmtime ($\alpha$=0.99, $\beta$=0.3) | 0.01 (0.00, 0.03) | 0.61 (0.51, 0.71) | 0.45 (0.35, 0.55) | 0.58 (0.48, 0.68) | 0.20 (0.12, 0.28) |
| llmtime ($\alpha$=0.99, $\beta$=0.5) | 0.01 (0.00, 0.03) | 0.51 (0.42, 0.60) | 0.47 (0.37, 0.57) | 0.61 (0.51, 0.70) | 0.24 (0.16, 0.33) |
| minmax | 0.00 (0.00, 0.00) | 0.57 (0.47, 0.66) | 0.59 (0.49, 0.67) | 0.67 (0.58, 0.76) | 0.20 (0.13, 0.28) |
| none | 0.17 (0.10, 0.25) | 0.92 (0.87, 0.97) | 0.48 (0.38, 0.58) | 0.79 (0.71, 0.87) | 0.29 (0.20, 0.37) |

Supplementary Table S11: Text-based functional form identification ablation results: rescaling values as suggested in (Gruver et al., 2024) for different function classes.

### A.3.3 PLOT-BASED RESULTS

| dpi | 50 | 500 | 1000 | 2500 |
|---|---|---|---|---|
| 25 | 0.66 (0.57, 0.74) | 0.67 (0.60, 0.75) | 0.70 (0.62, 0.78) | 0.66 (0.58, 0.74) |
| 50 | 0.59 (0.51, 0.67) | 0.70 (0.62, 0.78) | 0.70 (0.62, 0.78) | 0.74 (0.66, 0.82) |
| 100 | 0.67 (0.58, 0.75) | 0.70 (0.62, 0.78) | 0.68 (0.60, 0.76) | 0.74 (0.66, 0.81) |
| 200 | 0.72 (0.64, 0.80) | 0.70 (0.62, 0.78) | 0.72 (0.64, 0.80) | 0.73 (0.65, 0.81) |
| 400 | 0.63 (0.55, 0.71) | 0.71 (0.63, 0.79) | 0.71 (0.63, 0.79) | 0.73 (0.65, 0.81) |

Supplementary Table S12: Plot-based functional form identification ablation results: modifying figure dpi for different numbers of points.

| dpi | 0.0 | 0.5 | 1.0 | 2.0 | 5.0 |
|---|---|---|---|---|---|
| 25 | 0.91 (0.85, 0.96) | 0.83 (0.76, 0.90) | 0.61 (0.52, 0.71) | 0.61 (0.51, 0.70) | 0.41 (0.30, 0.51) |
| 50 | 0.92 (0.86, 0.97) | 0.87 (0.80, 0.93) | 0.68 (0.59, 0.77) | 0.59 (0.49, 0.69) | 0.35 (0.26, 0.44) |
| 100 | 0.95 (0.90, 0.99) | 0.89 (0.82, 0.95) | 0.65 (0.56, 0.74) | 0.60 (0.50, 0.70) | 0.40 (0.31, 0.50) |
| 200 | 0.98 (0.95, 1.00) | 0.89 (0.83, 0.95) | 0.69 (0.60, 0.78) | 0.61 (0.52, 0.70) | 0.42 (0.33, 0.52) |
| 400 | 0.94 (0.89, 0.98) | 0.89 (0.82, 0.95) | 0.68 (0.59, 0.77) | 0.58 (0.49, 0.68) | 0.39 (0.30, 0.49) |

Supplementary Table S13: Plot-based functional form identification ablation results: modifying figure dpi for different noise levels.

| dpi | cubic | exponential | linear | periodic | quadratic |
|---|---|---|---|---|---|
| 25 | 0.22 (0.14, 0.30) | 0.95 (0.91, 0.99) | 0.99 (0.97, 1.00) | 0.37 (0.28, 0.47) | 0.84 (0.77, 0.91) |
| 50 | 0.38 (0.29, 0.47) | 0.95 (0.91, 0.99) | 0.98 (0.95, 1.00) | 0.32 (0.23, 0.41) | 0.78 (0.70, 0.85) |
| 100 | 0.37 (0.28, 0.47) | 0.96 (0.92, 0.99) | 0.97 (0.93, 1.00) | 0.45 (0.35, 0.55) | 0.74 (0.65, 0.82) |
| 200 | 0.42 (0.32, 0.51) | 0.97 (0.93, 1.00) | 0.98 (0.95, 1.00) | 0.45 (0.36, 0.55) | 0.77 (0.68, 0.84) |
| 400 | 0.43 (0.33, 0.53) | 0.94 (0.89, 0.98) | 0.98 (0.95, 0.98) | 0.39 (0.29, 0.49) | 0.74 (0.65, 0.83) |

Supplementary Table S14: Plot-based functional form identification ablation results: modifying figure dpi for different function classes.

| figsize | 50 | 500 | 1000 | 2500 |
|---|---|---|---|---|
| (4, 3) | 0.67 (0.59, 0.75) | 0.70 (0.62, 0.78) | 0.71 (0.63, 0.79) | 0.71 (0.63, 0.79) |
| (3.5, 3.5) | 0.71 (0.63, 0.79) | 0.73 (0.65, 0.81) | 0.76 (0.68, 0.83) | 0.77 (0.70, 0.83) |
| (6.4, 4.8) | 0.69 (0.60, 0.76) | 0.72 (0.65, 0.80) | 0.70 (0.63, 0.78) | 0.72 (0.65, 0.80) |
| (8, 6) | 0.65 (0.57, 0.74) | 0.70 (0.62, 0.78) | 0.69 (0.60, 0.77) | 0.72 (0.65, 0.79) |
| (7, 7) | 0.63 (0.55, 0.72) | 0.74 (0.66, 0.81) | 0.71 (0.63, 0.79) | 0.75 (0.67, 0.82) |
| (12, 12) | 0.70 (0.62, 0.78) | 0.66 (0.58, 0.74) | 0.68 (0.60, 0.75) | 0.71 (0.63, 0.79) |

Supplementary Table S15: Plot-based functional form identification ablation results: modifying figure size for different numbers of points.

| figsize | 0.0 | 0.5 | 1.0 | 2.0 | 5.0 |
|---|---|---|---|---|---|
| (4, 3) | 0.98 (0.95, 1.00) | 0.86 (0.79, 0.93) | 0.65 (0.55, 0.75) | 0.63 (0.54, 0.72) | 0.37 (0.28, 0.47) |
| (3.5, 3.5) | 0.98 (0.95, 1.00) | 0.90 (0.83, 0.95) | 0.76 (0.68, 0.84) | 0.69 (0.60, 0.77) | 0.38 (0.29, 0.48) |
| (6.4, 4.8) | 0.97 (0.93, 1.00) | 0.89 (0.83, 0.95) | 0.66 (0.57, 0.75) | 0.59 (0.49, 0.69) | 0.43 (0.34, 0.53) |
| (8, 6) | 0.95 (0.90, 0.99) | 0.88 (0.82, 0.94) | 0.67 (0.59, 0.75) | 0.59 (0.49, 0.68) | 0.35 (0.26, 0.44) |
| (7, 7) | 0.95 (0.90, 0.99) | 0.88 (0.81, 0.94) | 0.76 (0.68, 0.84) | 0.60 (0.51, 0.70) | 0.35 (0.26, 0.44) |
| (12, 12) | 0.94 (0.89, 0.98) | 0.79 (0.70, 0.87) | 0.69 (0.60, 0.78) | 0.63 (0.54, 0.72) | 0.38 (0.29, 0.48) |

Supplementary Table S16: Plot-based functional form identification ablation results: modifying figure size for different noise levels.

| figsize | cubic | exponential | linear | periodic | quadratic |
|---|---|---|---|---|---|
| (4, 3) | 0.35 (0.26, 0.44) | 0.95 (0.90, 0.99) | 0.99 (0.97, 1.00) | 0.39 (0.29, 0.48) | 0.81 (0.73, 0.89) |
| (3.5, 3.5) | 0.57 (0.47, 0.66) | 0.96 (0.92, 0.99) | 0.99 (0.97, 1.00) | 0.36 (0.27, 0.46) | 0.83 (0.76, 0.90) |
| (6.4, 4.8) | 0.37 (0.28, 0.46) | 0.96 (0.92, 0.99) | 0.99 (0.96, 1.00) | 0.48 (0.38, 0.57) | 0.74 (0.65, 0.83) |
| (8, 6) | 0.40 (0.30, 0.50) | 0.94 (0.89, 0.98) | 0.97 (0.93, 1.00) | 0.39 (0.30, 0.49) | 0.74 (0.66, 0.83) |
| (7, 7) | 0.51 (0.41, 0.61) | 0.93 (0.88, 0.98) | 0.95 (0.90, 0.99) | 0.35 (0.26, 0.44) | 0.80 (0.72, 0.88) |
| (12, 12) | 0.31 (0.22, 0.40) | 0.91 (0.85, 0.96) | 1.00 (1.00, 1.00) | 0.43 (0.33, 0.53) | 0.78 (0.70, 0.85) |

Supplementary Table S17: Plot-based functional form identification ablation results: modifying figure size for different function classes.

| Plot style | 50 | 500 | 1000 | 2500 |
|---|---|---|---|---|
| None | 0.66 (0.57, 0.74) | 0.72 (0.64, 0.79) | 0.70 (0.62, 0.78) | 0.72 (0.64, 0.79) |
| classic | 0.70 (0.62, 0.78) | 0.72 (0.64, 0.79) | 0.70 (0.62, 0.78) | 0.73 (0.65, 0.80) |
| ggplot | 0.65 (0.56, 0.73) | 0.70 (0.62, 0.79) | 0.70 (0.62, 0.78) | 0.74 (0.67, 0.82) |
| seaborn-v0_8-darkgrid | 0.68 (0.60, 0.76) | 0.71 (0.63, 0.78) | 0.69 (0.60, 0.77) | 0.74 (0.66, 0.82) |
| seaborn-v0_8-whitegrid | 0.66 (0.58, 0.74) | 0.70 (0.62, 0.78) | 0.70 (0.62, 0.78) | 0.72 (0.63, 0.80) |

Supplementary Table S18: Plot-based functional form identification ablation results: modifying plotting style for different numbers of points.

| Plot style | 0.0 | 0.5 | 1.0 | 2.0 | 5.0 |
|---|---|---|---|---|---|
| None | 0.95 (0.91, 0.99) | 0.92 (0.86, 0.97) | 0.65 (0.55, 0.74) | 0.56 (0.47, 0.65) | 0.41 (0.31, 0.50) |
| classic | 0.96 (0.92, 0.99) | 0.91 (0.85, 0.96) | 0.67 (0.57, 0.76) | 0.59 (0.50, 0.68) | 0.43 (0.33, 0.52) |
| ggplot | 0.94 (0.89, 0.98) | 0.91 (0.85, 0.96) | 0.68 (0.59, 0.77) | 0.60 (0.50, 0.70) | 0.37 (0.28, 0.46) |
| seaborn-v0_8-darkgrid | 0.96 (0.92, 0.99) | 0.87 (0.80, 0.93) | 0.67 (0.57, 0.76) | 0.61 (0.51, 0.70) | 0.41 (0.32, 0.51) |
| seaborn-v0_8-whitegrid | 0.96 (0.92, 0.99) | 0.87 (0.81, 0.94) | 0.67 (0.58, 0.75) | 0.58 (0.48, 0.68) | 0.41 (0.32, 0.51) |

Supplementary Table S19: Plot-based functional form identification ablation results: modifying plotting style for different noise levels.

| Plot style | cubic | exponential | linear | periodic | quadratic |
|---|---|---|---|---|---|
| None | 0.39 (0.29, 0.48) | 0.97 (0.93, 1.00) | 0.97 (0.93, 1.00) | 0.43 (0.34, 0.53) | 0.73 (0.64, 0.82) |
| classic | 0.41 (0.32, 0.51) | 0.96 (0.92, 0.99) | 0.98 (0.95, 1.00) | 0.46 (0.37, 0.56) | 0.75 (0.67, 0.83) |
| ggplot | 0.42 (0.32, 0.51) | 0.94 (0.89, 0.98) | 0.96 (0.92, 0.99) | 0.43 (0.34, 0.53) | 0.75 (0.67, 0.84) |
| seaborn-v0_8-darkgrid | 0.36 (0.26, 0.45) | 0.95 (0.91, 0.99) | 1.00 (1.00, 1.00) | 0.46 (0.36, 0.55) | 0.75 (0.65, 0.84) |
| seaborn-v0_8-whitegrid | 0.35 (0.26, 0.45) | 0.97 (0.93, 1.00) | 0.97 (0.93, 1.00) | 0.47 (0.38, 0.57) | 0.73 (0.64, 0.82) |

Supplementary Table S20: Plot-based functional form identification ablation results: modifying plotting styles for different function classes.

| Color palette | 50 | 500 | 2500 |
|---|---|---|---|
| black and white | 0.50 (0.48, 0.51) | 0.64 (0.63, 0.65) | 0.67 (0.66, 0.69) |
| default | 0.50 (0.49, 0.52) | 0.63 (0.61, 0.64) | 0.65 (0.64, 0.67) |
| high contrast | 0.49 (0.48, 0.51) | 0.60 (0.58, 0.61) | 0.63 (0.61, 0.64) |
| invert | 0.51 (0.50, 0.53) | 0.64 (0.62, 0.65) | 0.66 (0.65, 0.68) |
| low contrast | 0.48 (0.47, 0.50) | 0.61 (0.60, 0.63) | 0.65 (0.63, 0.66) |

Supplementary Table S21: Plot-based functional form identification ablation results: modifying color palette for different numbers of points.

| Color palette | 0.0 | 1.0 | 5.0 |
|---|---|---|---|
| black and white | 0.88 (0.87, 0.89) | 0.63 (0.62, 0.65) | 0.30 (0.29, 0.32) |
| default | 0.88 (0.87, 0.89) | 0.63 (0.62, 0.65) | 0.27 (0.26, 0.29) |
| high contrast | 0.85 (0.84, 0.86) | 0.62 (0.60, 0.63) | 0.25 (0.24, 0.26) |
| invert | 0.88 (0.87, 0.89) | 0.64 (0.62, 0.65) | 0.30 (0.28, 0.31) |
| low contrast | 0.86 (0.85, 0.87) | 0.61 (0.60, 0.63) | 0.27 (0.25, 0.28) |

Supplementary Table S22: Plot-based functional form identification ablation results: modifying color palette for different noise levels.

| Color palette | cubic | exponential | linear | periodic | quadratic |
|---|---|---|---|---|---|
| black and white | 0.35 (0.33, 0.37) | 0.78 (0.76, 0.79) | 0.90 (0.88, 0.91) | 0.30 (0.28, 0.32) | 0.71 (0.69, 0.72) |
| default | 0.32 (0.30, 0.34) | 0.76 (0.74, 0.77) | 0.90 (0.89, 0.91) | 0.31 (0.29, 0.33) | 0.69 (0.67, 0.71) |
| high contrast | 0.27 (0.25, 0.28) | 0.73 (0.71, 0.74) | 0.90 (0.89, 0.91) | 0.28 (0.26, 0.30) | 0.69 (0.68, 0.71) |
| invert | 0.32 (0.30, 0.34) | 0.79 (0.77, 0.80) | 0.90 (0.89, 0.91) | 0.31 (0.29, 0.33) | 0.70 (0.69, 0.72) |
| low contrast | 0.29 (0.28, 0.31) | 0.75 (0.74, 0.77) | 0.88 (0.87, 0.89) | 0.29 (0.27, 0.30) | 0.69 (0.67, 0.71) |

Supplementary Table S23: Plot-based functional form identification ablation results: modifying color palette for different function classes.

| Plot marker | 50 | 500 | 2500 |
|---|---|---|---|
| + | 0.48 (0.47, 0.49) | 0.61 (0.60, 0.63) | 0.64 (0.63, 0.66) |
| ^ | 0.50 (0.49, 0.52) | 0.64 (0.63, 0.66) | 0.67 (0.66, 0.69) |
| o | 0.52 (0.51, 0.54) | 0.63 (0.62, 0.65) | 0.65 (0.64, 0.67) |
| s | 0.50 (0.48, 0.52) | 0.62 (0.60, 0.64) | 0.64 (0.63, 0.66) |
| x | 0.49 (0.47, 0.50) | 0.61 (0.59, 0.62) | 0.65 (0.64, 0.67) |

Supplementary Table S24: Plot-based functional form identification ablation results: modifying plot makers for different numbers of points.

| Plot marker | 0.0 | 1.0 | 5.0 |
|---|---|---|---|
| + | 0.86 (0.85, 0.87) | 0.62 (0.60, 0.63) | 0.26 (0.25, 0.27) |
| ^ | 0.88 (0.87, 0.89) | 0.64 (0.63, 0.66) | 0.30 (0.29, 0.32) |
| o | 0.88 (0.87, 0.89) | 0.63 (0.62, 0.65) | 0.29 (0.28, 0.31) |
| s | 0.88 (0.87, 0.89) | 0.62 (0.60, 0.63) | 0.27 (0.26, 0.28) |
| x | 0.86 (0.85, 0.87) | 0.62 (0.61, 0.64) | 0.27 (0.25, 0.28) |

Supplementary Table S25: Plot-based functional form identification ablation results: modifying plot makers for different noise levels.

| Plot marker | cubic | exponential | linear | periodic | quadratic |
|---|---|---|---|---|---|
| + | 0.29 (0.27, 0.31) | 0.75 (0.74, 0.77) | 0.89 (0.88, 0.90) | 0.28 (0.26, 0.30) | 0.68 (0.66, 0.69) |
| ^ | 0.31 (0.29, 0.33) | 0.77 (0.75, 0.78) | 0.89 (0.87, 0.90) | 0.34 (0.33, 0.36) | 0.73 (0.71, 0.74) |
| o | 0.32 (0.30, 0.34) | 0.78 (0.77, 0.80) | 0.90 (0.89, 0.91) | 0.30 (0.28, 0.32) | 0.71 (0.69, 0.73) |
| s | 0.33 (0.31, 0.35) | 0.75 (0.74, 0.77) | 0.91 (0.90, 0.92) | 0.28 (0.26, 0.29) | 0.68 (0.66, 0.70) |
| x | 0.30 (0.29, 0.32) | 0.75 (0.73, 0.76) | 0.89 (0.87, 0.90) | 0.28 (0.26, 0.30) | 0.70 (0.68, 0.72) |

Supplementary Table S26: Plot-based functional form identification ablation results: modifying plot makers for different function classes.

| Markers size | 50 | 500 | 2500 |
|---|---|---|---|
| large | 0.51 (0.49, 0.52) | 0.62 (0.61, 0.63) | 0.65 (0.64, 0.66) |
| medium | 0.50 (0.49, 0.51) | 0.62 (0.61, 0.63) | 0.65 (0.64, 0.66) |
| small | 0.49 (0.48, 0.50) | 0.63 (0.62, 0.64) | 0.65 (0.64, 0.66) |

Supplementary Table S27: Plot-based functional form identification ablation results: modifying plot maker sizes for different numbers of points.

| Marker size | 0.0 | 1.0 | 5.0 |
|---|---|---|---|
| large | 0.87 (0.86, 0.88) | 0.63 (0.62, 0.64) | 0.27 (0.26, 0.28) |
| medium | 0.87 (0.86, 0.88) | 0.63 (0.61, 0.64) | 0.28 (0.27, 0.29) |
| small | 0.87 (0.86, 0.87) | 0.62 (0.61, 0.63) | 0.28 (0.27, 0.29) |

Supplementary Table S28: Plot-based functional form identification ablation results: modifying plot maker sizes for different noise levels.

| Marker size | cubic | exponential | linear | periodic | quadratic |
|---|---|---|---|---|---|
| large | 0.32 (0.30, 0.33) | 0.75 (0.73, 0.76) | 0.91 (0.90, 0.92) | 0.30 (0.28, 0.31) | 0.70 (0.68, 0.71) |
| medium | 0.31 (0.30, 0.33) | 0.76 (0.75, 0.77) | 0.90 (0.89, 0.91) | 0.29 (0.27, 0.30) | 0.70 (0.69, 0.72) |
| small | 0.30 (0.29, 0.32) | 0.78 (0.76, 0.79) | 0.88 (0.87, 0.89) | 0.30 (0.29, 0.32) | 0.69 (0.68, 0.71) |

Supplementary Table S29: Plot-based functional form identification ablation results: modifying plot maker sizes for different function classes.

| Plot components | 50 | 500 | 2500 |
|---|---|---|---|
| all | 0.53 (0.51, 0.54) | 0.65 (0.64, 0.66) | 0.68 (0.67, 0.69) |
| minimal | 0.47 (0.46, 0.48) | 0.61 (0.60, 0.62) | 0.63 (0.62, 0.64) |
| none | 0.50 (0.49, 0.51) | 0.61 (0.60, 0.62) | 0.66 (0.64, 0.67) |

Supplementary Table S30: Plot-based functional form identification ablation results: modifying plot components for different numbers of points.

| Plot components | 0.0 | 1.0 | 5.0 |
| --- | --- | --- | --- |
| all | 0.89 (0.88, 0.90) | 0.65 (0.63, 0.66) | 0.32 (0.31, 0.33) |
| minimal | 0.83 (0.82, 0.84) | 0.60 (0.59, 0.61) | 0.28 (0.27, 0.29) |
| none | 0.89 (0.88, 0.90) | 0.63 (0.62, 0.65) | 0.24 (0.23, 0.25) |

Supplementary Table S31: Plot-based functional form identification ablation results: modifying plot components for different noise levels.

| Plot components | cubic | exponential | linear | periodic | quadratic |
| --- | --- | --- | --- | --- | --- |
| all | 0.37 (0.36, 0.39) | 0.82 (0.81, 0.83) | 0.94 (0.94, 0.95) | 0.29 (0.28, 0.31) | 0.66 (0.65, 0.68) |
| minimal | 0.24 (0.23, 0.25) | 0.75 (0.74, 0.76) | 0.89 (0.88, 0.90) | 0.25 (0.24, 0.27) | 0.71 (0.69, 0.72) |
| none | 0.32 (0.30, 0.33) | 0.71 (0.70, 0.72) | 0.85 (0.84, 0.86) | 0.34 (0.33, 0.35) | 0.72 (0.71, 0.74) |

Supplementary Table S32: Plot-based functional form identification ablation results: modifying plot components for different function classes.

| Temperature | 50 | 500 | 1000 | 2500 |
| --- | --- | --- | --- | --- |
| 0.00 | 0.70 (0.62, 0.77) | 0.71 (0.63, 0.79) | 0.68 (0.60, 0.75) | 0.72 (0.64, 0.80) |
| 0.10 | 0.66 (0.58, 0.74) | 0.71 (0.63, 0.78) | 0.70 (0.62, 0.78) | 0.73 (0.65, 0.80) |
| 0.30 | 0.72 (0.64, 0.80) | 0.72 (0.64, 0.80) | 0.68 (0.60, 0.77) | 0.73 (0.65, 0.80) |
| 0.55 | 0.69 (0.60, 0.78) | 0.70 (0.62, 0.78) | 0.70 (0.62, 0.77) | 0.74 (0.66, 0.82) |
| 1.00 | 0.71 (0.64, 0.78) | 0.70 (0.62, 0.78) | 0.68 (0.59, 0.76) | 0.74 (0.66, 0.81) |

Supplementary Table S33: Plot-based functional form identification ablation results: modifying temperature for different numbers of points.

| Temperature | 0.0 | 0.5 | 1.0 | 2.0 | 5.0 |
| --- | --- | --- | --- | --- | --- |
| 0.00 | 0.96 (0.92, 0.99) | 0.90 (0.84, 0.96) | 0.67 (0.58, 0.75) | 0.58 (0.47, 0.67) | 0.40 (0.31, 0.50) |
| 0.10 | 0.96 (0.92, 0.99) | 0.90 (0.84, 0.96) | 0.67 (0.58, 0.76) | 0.57 (0.48, 0.66) | 0.40 (0.31, 0.51) |
| 0.30 | 0.96 (0.91, 0.99) | 0.93 (0.87, 0.98) | 0.67 (0.58, 0.76) | 0.62 (0.53, 0.71) | 0.38 (0.29, 0.48) |
| 0.55 | 0.95 (0.90, 0.99) | 0.90 (0.84, 0.95) | 0.69 (0.60, 0.78) | 0.59 (0.49, 0.69) | 0.41 (0.32, 0.51) |
| 1.00 | 0.96 (0.92, 0.99) | 0.91 (0.85, 0.96) | 0.68 (0.58, 0.77) | 0.61 (0.52, 0.70) | 0.38 (0.28, 0.47) |

Supplementary Table S34: Plot-based functional form identification ablation results: modifying temperature for different noise levels.

| Temperature | cubic | exponential | linear | periodic | quadratic |
|---|---|---|---|---|---|
| 0.0 | 0.37 (0.27, 0.46) | 0.95 (0.91, 0.99) | 0.97 (0.93, 1.00) | 0.48 (0.38, 0.58) | 0.74 (0.65, 0.82) |
| 0.1 | 0.40 (0.30, 0.50) | 0.94 (0.89, 0.98) | 0.97 (0.93, 1.00) | 0.45 (0.35, 0.54) | 0.74 (0.66, 0.83) |
| 0.3 | 0.42 (0.33, 0.52) | 0.92 (0.86, 0.97) | 0.99 (0.97, 1.00) | 0.47 (0.38, 0.57) | 0.76 (0.67, 0.84) |
| 0.55 | 0.39 (0.30, 0.49) | 0.96 (0.92, 0.99) | 0.99 (0.97, 1.00) | 0.46 (0.37, 0.56) | 0.74 (0.65, 0.82) |
| 1.0 | 0.37 (0.28, 0.46) | 0.96 (0.91, 0.99) | 0.96 (0.92, 0.99) | 0.48 (0.38, 0.58) | 0.77 (0.69, 0.85) |

Supplementary Table S35: Plot-based functional form identification ablation results: modifying temperature for different function classes.

### A.3.4 Combined modality results

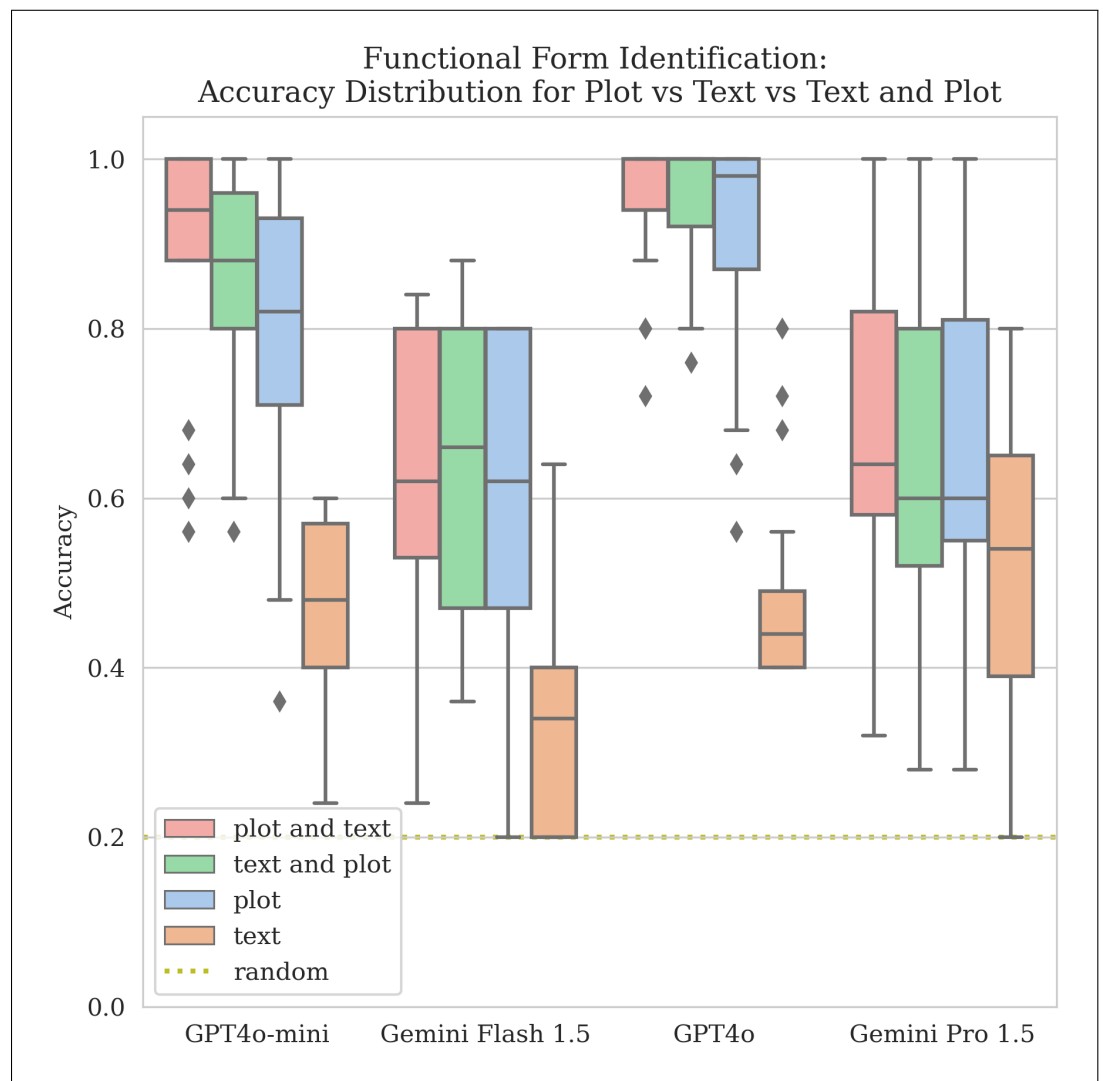

Supplementary Figure S15: Results of functional form identification task across all modalities, including combined plot and text.

## A.4 Prompts and target dataclasses

For reproducibility, we provide here the prompt templates and target dataclasses we provided to Langfun for each task, except for readiness as that was performed on a proprietary dataset.

### A.4.1 Functional form identification

```
FunctionType = typing.Literal[
    'exponential', 'periodic', 'quadratic', 'linear', 'cubic'
]

@dataclasses.dataclass
class FunctionClassification:
  reason: str
```

```
function_type: FunctionType
```

Target Dataclass

```
You are a professional data scientist with a great intuition for
looking for trends in data. You are taking your next exam
    which has a choose-the-correct-answer format. Answer the
    question below to the best of your ability.
    To maximise your score on the exam ALWAYS PROVIDE A BEST GUESS
    , even if you are not sure.

    Q: Classify the trend in the plot shown into one of the
    following types. Note that a periodic function can be sine,
     cosine, etc.
        {%- for f in function_type %}
          {{ f }}
        {%- endfor %}

        {{ plot }}

        The data may be noisy, so do your best to see the
        underlying trend. Provide a reason for your answer.
```

Plot Prompt

```
You are a professional data scientist with a great intuition for
looking for trends in data. You are taking your next exam
    which has a choose-the-correct-answer format. Answer the
    question below to the best of your ability.
    To maximise your score on the exam ALWAYS PROVIDE A BEST GUESS
    , even if you are not sure.

    Q: Classify the trend in the data shown into one of the
    following types. Note that a periodic function can be sine,
     cosine, etc.
        {%- for f in function_type %}
          {{ f }}
        {%- endfor %}

        x: {{ x }}
        y: {{ y }}

        The data may be noisy, so do your best to see the
        underlying trend. Provide a reason for your answer.
```

Text Prompt

```
You are a professional data scientist with a great intuition for
looking for trends in data. You are taking your next exam
    which has a choose-the-correct-answer format. Answer the
    question below to the best of your ability.
    To maximise your score on the exam ALWAYS PROVIDE A BEST GUESS
    , even if you are not sure.

    Q: Classify the trend in the data shown into one of the
    following types. Note that a periodic function can be sine,
     cosine, etc.
        {%- for f in function_type %}
          {{ f }}
```

```
        {%- endfor %}

    Here are the data presented as lists of values:
        x: {{ x }}
        y: {{ y }}

    Here are the same data presented as a plot:
        {{ plot }}

    The data may be noisy, so do your best to see the
    underlying trend. Provide a reason for your answer.
```

Text and Plot Prompt

```
You are a professional data scientist with a great intuition for
looking for trends in data. You are taking your next exam
    which has a choose-the-correct-answer format. Answer the
    question below to the best of your ability.
    To maximise your score on the exam ALWAYS PROVIDE A BEST GUESS
    , even if you are not sure.

    Q: Classify the trend in the data shown into one of the
    following types. Note that a periodic function can be sine,
     cosine, etc.
        {%- for f in function_type %}
          {{ f }}
        {%- endfor %}

    Here are the same data presented as a plot:
        {{ plot }}

    Here are the data presented as lists of values:
        x: {{ x }}
        y: {{ y }}

    The data may be noisy, so do your best to see the
    underlying trend. Provide a reason for your answer.
```

Plot and Text Prompt

### A.4.2 CORRELATION OF TWO LINES

```
CorrelationType = typing.Literal[
    'Positively␣Correlated', 'Negatively␣Correlated'
]

@dataclasses.dataclass
class FunctionCorrelationResult:
  reason: str
  correlation_type: CorrelationType
```

Target Dataclass

```
##############################################################
######################
You are a professional data scientist with a great intuition for
looking for trends in data.
```

```
Answer the question below to the best of your ability.
The data might be noisy.
Provide reasoning.
######################################################################
#######################

Q: Observe two plots y1=f(x) and y2=g(x) over the same range of x.

plot: {{ plot }}

Find out if they are positively or negatively correlated. Provide
your reasoning.

You may want to follow the following steps to solve this problem:
Analyze the two functions and find out when one is increasing
whether the other one is decreasing.
If both functions tend to increase and decrease together, they are
 positively correlated.
If one function tends to increase and the other one tends to
decrease, they are negatively correlated.
```

Plot Prompt

```
######################################################################
#######################
You are a professional data scientist with a great intuition for
looking for trends in data.
Answer the question below to the best of your ability.
The data might be noisy.
Provide reasoning.
######################################################################
#######################

Q: Analyze two functions y1=f(x) and y2=g(x) defined over the same
 range of x.

x: {{ x }}
y1: {{ y1 }}
y2: {{ y2 }}

Find out if they are positively or negatively correlated. Provide
your reasoning.

You may want to follow the following steps to solve this problem:
Analyze the two functions and find out when one is increasing
whether the other one is decreasing.
If both functions tend to increase and decrease together, they are
 positively correlated.
If one function tends to increase and the other one tends to
decrease, they are negatively correlated.
```

Text Prompt

### A.4.3  2D CLUSTER COUNTING

```
@dataclasses.dataclass
class ClusterCount:
  cluster_count: int
```

```
  reason: str
```
Target Dataclass

```
Q: Here is a scatter plot of data points. The points form radial
clusters. The cluster count can be 1, 2, 3, 4, 5, 6, 7, 8, or 9.
Your task is to count the number of clusters from this plot.
As an intermediate step, estimate the positions of the cluster
centers, followed by the number of clusters.

{{ plot }}

A:
```
Plot Prompt

```
Q: Count the number of clusters. The possible number of clusters
is 1, 2, 3, 4, 5, 6, 7, 8, or 9. The clusters are radial in shape.
As an intermediate step, estimate the positions of the cluster
centers.

Here are the data points. Only provide the cluster count, a rough
estimate is fine too. The data may be noisy, just make your best
guess, no explanations needed.
x: {{ x }}
y: {{ y }}

A:
```
Text Prompt

### A.4.4 DERIVATIVE IDENTIFICATION

```
MCQChoice = typing.Literal['1', '2', '3', '4']

@dataclasses.dataclass
class DerivativeMCQChoice:
  reason: str
  mcq_choice: MCQChoice
```
Target Dataclass

```
#####################################################################
#######################
You are a professional data scientist with a great intuition for
looking for trends in data.
You are taking your next exam which has a choose-the-correct-
answer format.
Answer the question below to the best of your ability.
To maximise your score on the exam ALWAYS PROVIDE A BEST GUESS,
even if you are not sure.
#####################################################################
#######################
```

```
Q: Observe the trend of the first plot below. Now, based on the
trend, select one of the
following plots that corresponds to the derivative of the original
 plot.

Provide your reasoning, and then return an answer. Your reasoning
should
include a description of the trend of the original plot, the
description of the trends of all the choices (1-4), and then
careful reasoning to select the correct answer. Please find the
plots
below.

Original plot:
{{ plots[0] }}

{%- for plot in plots[1:] %}
  Choice: {{ loop.index }}
  {{ plot }}
{%- endfor %}
```

Plot Prompt

```
######################################################################
#######################
You are a professional data scientist with a great intuition for
looking for trends in data.
You are taking your next exam which has a choose-the-correct-
answer format.
Answer the question below to the best of your ability.
To maximise your score on the exam ALWAYS PROVIDE A BEST GUESS,
even if you are not sure.
######################################################################
#######################

Q: Consider the first set of data provided as lists of x and y
points, and
consider its trend.

The next four sets of data are labelled 1, 2, 3, 4 based on the
order in
which I give them to you, and represent potential derivatives as
lists of x
and dy points. Of these, choose the dataset number (1, 2, 3 or 4)
that
corresponds to the derivative of the original data. Provide your
reasoning
before your answer.

    Original data:
        x: {{ x }}
        y: {{ y }}

    {%- for d in derivatives %}
    Dataset {{ loop.index }}:
        x: {{ d[0] }}
        dy: {{ d[1] }}
    {%- endfor %}
```

---

Text Prompt

### A.4.5 QUADRATIC DERIVATIVE IDENTIFICATION

```
MCQChoice = typing.Literal['1', '2', '3', '4']

@dataclasses.dataclass
class QuadraticDerivativeMCQChoice:
  reason: str
  mcq_choice: MCQChoice
```

Target Dataclass

```
####################################################################
#######################
You are a professional data scientist with a great intuition for
looking for trends in data.
You are taking your next exam which has a choose-the-correct-
answer format.
Answer the question below to the best of your ability.
To maximise your score on the exam ALWAYS PROVIDE A BEST GUESS,
even if you are not sure.
####################################################################
#######################

Q: Observe the slope and magnitude of the first plot of a
quadratic function
below. Now, based on both slope and magnitude, select one of the
following
plots that corresponds to the derivative of the original plot.
Note that
many of the choices might have the same slope sign, so you will
have to also
consider the magnitude of the y-values to get a correct answer.

The following reasoning will help you choose the right answer:
- From the shape of the original data, determine the function
class, and
thus the function class of the derivative.
- From the shape of the original data, determine the trend of the
expected
derivative, and thus the sign of its parameters.
- Use a few points from the original data to determine the
mangitude of the
original function's parameters, and thus the magnitude of the
expected
derivative's parameters.
- For each possible derivative choice, consider the shape of the
derivative and therefrom the sign of its parameters.  Also use a
few points
to determine the magnitude of the derivative choice's parameters.
- Compare the trend, sign and magnitudes of each derivative choice
 with the
expected result from observing the original data, and choose the
answer that
```

```
best matches.

Provide your reasoning, and then return an answer. Your reasoning
should
include a description of the slope and magnitude of the original
plot, the
description of the slope and magnitudes of all the choices (1-4),
and then
careful reasoning to select the correct answer. Please find the
plots
below.

Original plot:
{{ plots[0] }}

Choices below

{%- for plot in plots[1:] %}
Choice: {{ loop.index }}
   {{ plot }}
{%- endfor %}
```

Plot Prompt - zero-shot

```
######################################################################
#######################
You are a professional data scientist with a great intuition for
looking for trends in data.
You are taking your next exam which has a choose-the-correct-
answer format.
Answer the question below to the best of your ability.
To maximise your score on the exam ALWAYS PROVIDE A BEST GUESS,
even if you are not sure.
######################################################################
#######################

Q: Observe the slope and magnitude of the first set of x and y
points
sampled from a potentially noisy quadratic function below. Now,
based on
both slope and magnitude, select one of the following choices of x
 and y
points that corresponds to the derivative of the original data.
Note that
many of the choices might have the same slope sign, so you will
have to also
consider the magnitude of the y-values to get a correct answer.
The following reasoning will help you choose the right answer:
- From the shape of the original data, determine the function
class, and
thus the function class of the derivative.
- From the shape of the original data, determine the trend of the
expected
derivative, and thus the sign of its parameters.
- Use a few points from the original data to determine the
mangitude of the
original function's parameters, and thus the magnitude of the
expected
```

```
derivative's parameters.
- For each possible derivative choice, consider the shape of the
derivative and thereform the sign of its parameters.  Also use a
few points
to determine the magnitude of the derivative choice's parameters.
- Compare the trend, sign and magnitudes of each derivative choice
 with the
expected result from observing the original data, and choose the
answer that
best matches.

Provide your reasoning, and then
return an answer. Your reasoning should include a description of
the slope
and magnitude of the original data, the description of the slope
and
magnitudes of all the choices (1-4), and then careful reasoning to
 select
the correct answer. Please find the data below.

Original data:
    x: {{ x }}
    y: {{ y }}

Choices below

{%- for d in derivatives %}
Choice: {{ loop.index }}
    x: {{ d[0] }}
    dy: {{ d[1] }}
{%- endfor %}
```

Text Prompt - zero-shot

```
#####################################################################
#######################
You are a professional data scientist with a great intuition for
looking for trends in data.
You are taking your next exam which has a choose-the-correct-
answer format.
Answer the question below to the best of your ability.
To maximise your score on the exam ALWAYS PROVIDE A BEST GUESS,
even if you are not sure.
#####################################################################
#######################

Q: Observe the slope and magnitude of the first plot of a
quadratic function
below. Now, based on both slope and magnitude, select one of the
following
plots that corresponds to the derivative of the original plot.
Note that
many of the choices might have the same slope sign, so you will
have to also
consider the magnitude of the y-values to get a correct answer.

Here are some examples of how to approach this task:
```

```
{%- for example in fewshots %}

***** Example *****
  Original plot:
    {{ example.plots[0] }}

  Choices below:

  {%- for plot in example.plots[1:] %}
  Choice: {{ loop.index }}
    {{ plot }}
  {%- endfor %}

  Reasoning:
  I know that the original function is quadratic, therefore the
  derivative
  must be linear.

  I see that the original quadratic function opens
  {{ "up" if example.quadratic_scale > 0 else "down" }}, therefore
   the
  derivative must have a
  {{ "positive" if example.quadratic_scale > 0 else "negative" }}
  slope.

  Furthermore I see that the original function goes from a value
  of y=0
  around x=0 to a value of y={{ example.quadratic_scale }} around
  x=1,
  therefore the original function must be of the form
  y={{ example.quadratic_scale }} * x^2, therefore the derivative'
  s slope
  must have a magnitude of {{ 2 * example.quadratic_scale }}.

  Of the choices I've been given:
    {%- for scale in example.mcq_scales %}
    Choice {{ loop.index }}:
    - the line is {{ "increasing" if scale > 0 else "decreasing"
    }}, so the
    slope must be {{ "positive" if scale > 0 else "negative" }}
    - the line goes from having a value of y=0 around x=0 to a
    value of
    y={{ 2 * scale }} around x=1, so the value of the slope must
    be
    {{ 2 * scale}}
    - hence the derivative is a line with slope {{ 2 * scale}} and
    corresponds to an original quadratic function that opens
    {{ "up" if scale > 0 else "down" }}
    {%- endfor %}

  Only choice number {{ example.mcq_correct_idx_one_indexed }} has
   the
  correct direction and slope magnitude.

  Therefore the correct answer must be *choice number {{ example.
  mcq_correct_idx_one_indexed }}*.

{%- endfor %}
```

```
***** Your turn *****

Provide your reasoning, and then return an answer. Your reasoning
should
include a description of the slope and magnitude of the original
plot, the
description of the slope and magnitudes of all the choices (1-4),
and then
careful reasoning to select the correct answer. Please find the
plots
below.

Original plot:
{{ plots[0] }}

Choices below

{%- for plot in plots[1:] %}
Choice: {{ loop.index }}
  {{ plot }}
{%- endfor %}
```

Plot Prompt - few-shot

```
######################################################################
#######################
You are a professional data scientist with a great intuition for
looking for trends in data.
You are taking your next exam which has a choose-the-correct-
answer format.
Answer the question below to the best of your ability.
To maximise your score on the exam ALWAYS PROVIDE A BEST GUESS,
even if you are not sure.
######################################################################
#######################

Q: Observe the slope and magnitude of the first set of x and y
points
sampled from a potentially noisy quadratic function below. Now,
based on
both slope and magnitude, select one of the following choices of x
 and y
points that corresponds to the derivative of the original data.
Note that
many of the choices might have the same slope sign, so you will
have to also
consider the magnitude of the y-values to get a correct answer.

Here are some examples of how to approach this task:

{%- for example in fewshots %}

***** Example *****
  Original data:
      x: {{ example.x }}
      y: {{ example.x }}

  Choices below:
```

```
   {%- for d in example.derivatives %}
   Choice: {{ loop.index }}
       x: {{ d[0] }}
       dy: {{ d[1] }}
   {%- endfor %}

   Reasoning:
   I know that the original function is quadratic, therefore the
   derivative
   must be linear.

   I see that the original quadratic function opens
   {{ "up" if example.quadratic_scale > 0 else "down" }}, therefore
    the
   derivative must have a {{ "positive" if example.quadratic_scale
   > 0 else "negative" }} slope.

   Furthermore I see that the original function goes from a value
   of y=0
   around x=0 to a value of y={{ example.quadratic_scale }} around
   x=1,
   therefore the original function must be of the form
   y={{ example.quadratic_scale }} * x^2, therefore the derivative'
   s slope
   must have a magnitude of {{ 2 * example.quadratic_scale }}.

   Of the choices I've been given:
     {%- for scale in example.mcq_scales %}
     Choice {{ loop.index }}:
     - the line is {{ "increasing" if scale > 0 else "decreasing"
     }}, so the
     slope must be {{ "positive" if scale > 0 else "negative" }}
     - the line goes from having a value of y=0 around x=0 to a
     value of
     y={{ 2 * scale }} around x=1, so the value of the slope must
     be {{ 2 * scale}}
     - hence the derivative is a line with slope {{ 2 * scale}} and
     corresponds to an original quadratic function that opens {{ "
     up" if scale > 0 else "down" }}
     {%- endfor %}

   Only choice number {{ example.mcq_correct_idx_one_indexed }} has
    the
   correct direction and slope magnitude.

   Therefore the correct answer must be *choice number {{ example.
   mcq_correct_idx_one_indexed }}*.

{%- endfor %}

***** Your turn *****

Provide your reasoning, and then return an answer. Your reasoning
should
include a description of the slope and magnitude of the original
data, the
description of the slope and magnitudes of all the choices (1-4),
and then
```

```
careful reasoning to select the correct answer.

Please find the data below.

Original data:
    x: {{ x }}
    y: {{ y }}

Choices below

{%- for d in derivatives %}
Choice: {{ loop.index }}
    x: {{ d[0] }}
    dy: {{ d[1] }}
{%- endfor %}
```

Text Prompt - few-shot

### A.4.6 FALL DETECTION FROM IMU DATA

```
@dataclasses.dataclass
class Fall:
  fall_type: Literal["ADLs", "Falls", "Near"]
```

Target Dataclass

```
{%- for img, label in zip(few_shot_series, few_shot_labels) %}
Given that the following plot was classified as {{ label }}:
{{ img }}
{%- endfor %}
Classify the following plot in one of the following classes: ADLs,
 Falls, Near.
{{ sample }}
ALWAYS provide a best guess, since you will be graded on your
response.
```

Plot Prompt

```
{%- for data, label in zip(few_shot_series, few_shot_labels) %}
Given that the following time-series data was classified as {{
label }}:
{{ data }}
{%- endfor %}
Classify the following time-series data as 'ADLs', 'Falls', or '
Near':
{{ sample }}
ALWAYS provide a best guess, since you will be graded on your
response.
```

Text Prompt

### A.4.7 ACTIVITY RECOGNITION FROM IMU DATA

```
@dataclasses.dataclass
class ActivityNoUnknown:
  activity_type: Literal["bike", "sit", "stand", "walk", "stairs"]
```

**Target Dataclass**

```
{%- for imgs, label in zip(few_shot_series, few_shot_labels) %}
Given that the following plots were classified as {{ label }}:
{%- for img in imgs %}
{{ img }}
{%- endfor %}
{%- endfor %}
Classify the following plots in one of the following classes: {{
classes }}.
{%- for img in sample %}
{{ img }}
{%- endfor %}
ALWAYS provide a best guess, since you will be graded on your
response.
```

**Plot Prompt**

```
{%- for data, label in zip(few_shot_series, few_shot_labels) %}
Given that the following time-series data was classified as {{
label }}:
{{ data }}
{%- endfor %}
Classify the following time-series data in one of the following
classes: {{ classes }}.
{{ sample }}
ALWAYS provide a best guess, since you will be graded on your
response.
```

**Text Prompt**

