# OpenReview forum: "Plots unlock time-series understanding in multimodal models"
_ICLR.cc/2025/Conference — Submitted to ICLR 2025_

### Official Review · Reviewer_RmeP · 2024-10-28

**Soundness:** 3
**Presentation:** 3
**Contribution:** 3
**Rating:** 6
**Confidence:** 4

**Summary:**

The primary goal of the paper is to propose a novel yet practical method for enhancing time-series data analysis by using vision tokens and multimodal foundation models. This approach avoids the limitations inherent in processing long sequences of floating-point numbers by transforming them into visual representations (plots).  It showcases that the plot-based approach performs much better on tasks where understanding the overall trend is crucial, while also being more resource-efficient, highlighting a practical solution rather than a purely theoretical one.

**Strengths:**

The paper tackles how to enhance interpretability and efficiency when dealing with time-series data using multimodal models. It demonstrates examples of identifying clusters, recogniting real-world patterns, detecting, identifying the correlation between functions, and so on. This paper contains a wealth of comparative experiments, ablation studies on visual elements, and provides prompt templates in the supplementary material to ensure reproducibility. Overall, I appreciate the contribution of visual time-series representations and believe it add significant value to the ICLR community.

**Weaknesses:**

Lack of insights from a large number of experiments and ablation studies.

**Questions:**

1. How are multiple time series handled? Please add more details. For example, for the fall detection task on IMU data, is the input {img} a scatter plot containing six signals? Does a scatter plot containing multiple points from various variables lead to errors in the results? Including descriptions or figures about the input plots would enhance understanding.

2. Regarding the ablation experiments on visual elements in the matplotlib figure (including size, plot style, color palettes, markers, and plot components), which setting combination is reasonable or optimal? Is there a relationship between plot settings and the number of shown points? Since this paper focuses on plot representation, such a discussion is noteworthy and would be helpful for the follow-up work.

3. From the supplementary tables, it appears that results on cubic and periodic functions are worse on the plot-based approach. Some discussions on this finding are appreciated.

4. It might be beneficial to consider elaborating on the scalability of the approach for large-scale datasets.

---

> ### Author Response · Authors · 2024-11-27
> **Response to reviewer comments**
>
> Thanks very much for your thorough reading and helpful and positive feedback!
>
> **Regarding your specific questions:**
>
> > How are multiple time series handled? Please add more details. For example, for the fall detection task on IMU data, is the input {img} a scatter plot containing six signals? Does a scatter plot containing multiple points from various variables lead to errors in the results? Including descriptions or figures about the input plots would enhance understanding.
>
> We have expanded the details in the manuscript to add clarity around the shape of the input plots. Specifically, we included plotting code snippets for the tasks in Supplementary Section A.1 (Detailed Task Descriptions) starting on page 13, and examples of IMU data being plotted as done for the fall detection (with all 6 IMU dimensions plotted on one figure) in Supplementary Figure S6 (page 19)  activity recognition tasks (with 3 accelerometer and 3 gyroscope axes respectively plotted on separate figures) in Supplementary Figure S7 (page 21).  For Activity Recognition we did find that if we split the IMU data into two separate plots, with the 3 accelerometer axes and 3 gyroscope axes separately, there was a marginal performance gain vs plotting all 6 axes on the same plot; see new Supplementary Figure S8 on page 22.
>
> Regarding the number of points used, we subsampled down to 10 Hz for both datasets, using avgpool, due to GPT-4o context window constraints.
>
> > Regarding the ablation experiments on visual elements in the matplotlib figure (including size, plot style, color palettes, markers, and plot components), which setting combination is reasonable or optimal? Is there a relationship between plot settings and the number of shown points? Since this paper focuses on plot representation, such a discussion is noteworthy and would be helpful for the follow-up work.
>
> Thanks for the question.  We include the ablations described in Supplementary Information Section A.3 starting on page 28 to show that the plotting performance is generally robust to aesthetic choices, rather than as a rigorous investigation of the impacts, or optimisation, of these choices.
>
> The exact appearance of the time-series plots that lead to optimal performance would likely depend intimately on the exact downstream use or user requirement, so a priori cannot be determined. As such, we did not focus on carefully optimising the specific plotting approach, which could in theory be automated and forms the basis for future work – thank you for bringing this important point and we have highlighted it in the penultimate paragraph of the revised Section 5 (Conclusions and Future Work) on page 10.
>
> > From the supplementary tables, it appears that results on cubic and periodic functions are worse on the plot-based approach. Some discussions on this finding are appreciated.
>
> This is a good question – we would like to point to Supplementary Figure S9(c) on page 24 which illustrates the performances across models and modalities for different function classes for the function classification task.  Here we see that for all classes except periodic (including cubic), plots outperform or are within error bars of the text method, even if the absolute performances are lower than for other classes.  For the periodic functions, we found that the text models were biased towards this answer choice, so this class was a low precision (but high recall) answer.  As such, we believe the text models are not necessarily better at identifying periodic functions.
>
> > It might be beneficial to consider elaborating on the scalability of the approach for large-scale datasets.
>
> Thank you for raising this good point, which we have expanded on in the manuscript in the second paragraph of Section 4.4 (Cost) starting on page 9.  Assuming large-scale in the context of longer time-series, the visual representation approach we study here will scale better than a textual representation of the same time-series data. This is because the number of tokens required to represent the textual data scales roughly with the number of data points n, while for a fixed plot size the number of tokens required to represent an image is constant with respect to n.
>
> **To your other points of feedback:**
>
> > Lack of insights from a large number of experiments and ablation studies.
>
> As discussed above, our ablations (especially the plotting ones) are mainly included to show general robustness of the trend that visual representations outperform textual representations, rather than an attempt to optimise any one parameter to get the best absolute task performance out of the models.  As such, we felt that focussing too much discussion on the results of the ablation studies would distract from the main point at the cost of space better spent on what we feel are the more relevant details findings.  We appreciate your comment though, and would welcome any additional feedback on specific ablation areas you feel might remain unclear.

---

### Official Review · Reviewer_fw6B · 2024-10-28

**Soundness:** 1
**Presentation:** 1
**Contribution:** 1
**Rating:** 1
**Confidence:** 2

**Summary:**

This paper proposes a method for a multimodal foundation model to interpret time-series data by converting it into visual plots.
This approach employs the vision encoders of models like GPT-4 and Gemini to see visual time-series data instead of processing it as text, which, according to the authors, results in improved performance and reduced computational costs.
The paper evaluates the proposed method through synthetic and real-world tasks, including fall detection and activity recognition.

**Strengths:**

- The paper introduces a creative idea of using vision encoders to understand time-series data by visualizing it as plots.
- Multimodal models for time-series understanding through visualization potentially reduce the token usage, leading to lower API costs.
- The experiments cover both synthetic and real-world datasets, providing an assessment of the proposed method's strengths and weaknesses across different contexts.

**Weaknesses:**

- The core idea of using multimodal models to interpret time-series data via visual plots may not be sufficiently novel.
- The experimental methodology lacks rigorous justification. The paper does not provide enough comparative analysis with well-established baselines or detailed theoretical grounding to support its hypothesis, which weakens its scientific contribution.
- The related work section is not well-organized, making it difficult for readers to understand the context and significance of the contribution. This lack of structure makes it challenging to discern the novelty and value of the proposed method.
- The paper makes strong claims, such as the multimodal models' capability to mirror human understanding of visual plots, without adequate empirical backing or reference to previous research. Overstated claims with insufficient evidence can raise concerns about the validity of the conclusions.

**Questions:**

- Could you provide references or empirical evidence to support the claim that the model's approach mirrors the human approach?
- Could you consider grouping the related work into subsections based on different approaches to make it clearer?
- Could you add a clear mathematical formulation to define the problem and solution, which would make it more rigorous?
- Could you consider how to enhance the depth of analysis to meet the standards of high-quality research for ICLR?

---

> ### Author Response · Authors · 2024-11-27
> **Response to reviewer comments [1 / 2]**
>
> Thanks very much for your thorough reading and helpful feedback!
>
> **Regarding your specific questions:**
>
> > Could you provide references or empirical evidence to support the claim that the model's approach mirrors the human approach?
>
> Thank you for pointing out the gap in evidence related to this claim.  We have added the following citations to the first paragraph of Section 1 (Introduction) on page 1 to justify the claim that visual approaches can help humans ingest complex data:
> - S. K. Card, J. D. Mackinlay, and B. Shneiderman, “Readings in Information Visualization: Using Vision to Think,” San Francisco, CA, USA: Morgan Kaufmann Publishers Inc., 1999.
> - M. Adil Yalçin, Niklas Elmqvist, and Benjamin B. Bederson. 2016. Cognitive Stages in Visual Data Exploration. In Proceedings of the Sixth Workshop on Beyond Time and Errors on Novel Evaluation Methods for Visualization (BELIV '16). Association for Computing Machinery, New York, NY, USA, 86–95. https://doi.org/10.1145/2993901.2993902
>
> To be clear, our claim is that models benefit from using visual ingestion of complex data relative to textual representations similarly to how humans benefit from visual representations of data, not that the reasoning mechanism in which models parse visual data is the same as the cognitive mechanism with which humans comprehend visual stimuli.  We have included this important point in the first paragraph of Section 5 (Conclusions and Future Work) on page 10.
>
> > Could you consider grouping the related work into subsections based on different approaches to make it clearer?
>
> Thank you for this helpful suggestion – we have re-organised Section 2 (Related Work) on pages 2-3 into subsections based on approaches including “Forecasting”, “Time-series models”, “Vision models and visual representations” and “Measuring understanding".
>
> > Could you add a clear mathematical formulation to define the problem and solution, which would make it more rigorous?
>
> Our work is an empirical contribution, in which we seek to probe the capabilities of existing models without additional training, investigation of the internal machineries or novel encoding/decoding mechanisms.  As such, we don’t believe that there are relevant mathematical models to quote.  Furthermore, the black-box, closed-source nature of the models we study here preclude a closed-form expression of exactly how various inputs are tokenized, encoded, processed and decoded.  We also assume that the quality metrics (accuracy, mean absolute error, and F1-score) we employ are sufficiently familiar to the readership of ICLR as to warrant omission of definitions.
>
> > Could you consider how to enhance the depth of analysis to meet the standards of high-quality research for ICLR?
>
> We believe the analysis we employ here to be of the appropriate quality. We note that other reviewers (e.g. reviewer LLVU) have listed our analytical approach as a strength of our work, but would welcome any constructive feedback on the methodology as described in Section 3 on pages 3 and 4.

---

> > ### Author Response · Authors · 2024-11-27
> > **Response to reviewer comments [2 / 2]**
> >
> > **To your other points of feedback you brought:**
> >
> > > The core idea of using multimodal models to interpret time-series data via visual plots may not be sufficiently novel.
> >
> > Thank you for pointing out the lack of clarity in our message - we have updated our manuscript to emphasise that our core contribution is a rigorous evaluation of the differential performance between visual and textual encoding of time-series data across multimodal models, rather than a claimed novel proposition that models can reason on plotted time-series data at all. This is now in the last paragraph of Section 1 (Introduction) on page 2. To our knowledge, our work is the first in the field that empirically evaluates these two different ways in which a model might ingest time-series data.  As such, we believe that this novelty is an important contribution to understanding the general capabilities of multimodal foundation models.
> >
> > > The experimental methodology lacks rigorous justification. The paper does not provide enough comparative analysis with well-established baselines or detailed theoretical grounding to support its hypothesis, which weakens its scientific contribution.
> >
> > Related to the previous point, we emphasise that our work is the systematic investigation of the relative performances of visual versus textual encodings on these tasks across foundational models.  As such, the appropriate baseline, and the one which we employ, is the performance of a foundational model on the text representation of the data – we have added this point to the second paragraph of Section 3 on page 4. To put the absolute performances into context, we also provide random baselines to show that there is utility at all in using a foundation model in Figures 1-5, and provide state-of-the-art performance achieved by task-specific models for the fall detection and activity recognition real-world tasks in the caption of Figure 3, Figure 4 and Supplementary Table S1. To be clear, we are not claiming that our approach replaces or obviates training task-specific models, which we expect will achieve the highest absolute performance, but rather quantifying the differential ability of foundation models to ingest visual versus textual information.
> >
> > As mentioned, we also present our work as an empirical study of existing closed-source models, without a theoretical innovation.  As such, there is little additional mathematical or otherwise first-principles grounding to provide.
> >
> > The related work section is not well-organized, making it difficult for readers to understand the context and significance of the contribution. This lack of structure makes it challenging to discern the novelty and value of the proposed method.
> >
> > We appreciate this comment and have re-organised Section 2 (Related Works) on pages 2-3 so as to be more easily understood by readers.
> >
> > > The paper makes strong claims, such as the multimodal models' capability to mirror human understanding of visual plots, without adequate empirical backing or reference to previous research. Overstated claims with insufficient evidence can raise concerns about the validity of the conclusions.
> >
> > As described in an earlier point, we have added specific references to strengthen the claim referred to by the reviewer in the first paragraph of Section 1 (Introduction) on page 1. We believe that the other claims we make are well substantiated by a combination of our empirical findings and previous research, and highlight that other reviewers did not share this concern about unsubstantiated claims.  That said, we would very warmly welcome any other specific areas where you might feel that our claims are overreaching.

---

### Official Review · Reviewer_V8rN · 2024-10-28

**Soundness:** 2
**Presentation:** 2
**Contribution:** 2
**Rating:** 5
**Confidence:** 3

**Summary:**

Multimodal foundation models have a native capability to “see,” but this capability is currently underutilized. This paper introduces a method for leveraging the vision encoders of foundation models to interpret visual time-series data displayed in plots, allowing the foundation models to “see” the data; since LLMs are currently not well-suited for interpreting large sequences of numbers in time-series data, introducing the capability of visual interpretation can improve a model’s performance on a variety of time-series understanding tasks. These tasks include the ability to reason about trends, the relationship between multiple time-series, cluster data, and perform other time-series understanding tasks. The authors show that in both synthetic and real-world tasks, the plot performance improves substantially over text performance and is cheaper, with no additional model training.

**Strengths:**

1. The comparison to human interpretation of numerical versus visual input helped to motivate why this enables a new mode of reasoning in foundation models.
2. On some synthetic and real data tasks, visual interpretation of the data does lead to a notable improvement over text interpretation on many pattern-recognition tasks.

**Weaknesses:**

1. It seems like the actual task of recognizing the types of visual patterns achieved by this method are not inherently novel – Section 2 shows that models are already able to parse plots and tables. So the contribution of the paper would be more clear if the authors provided (1) comparisons to actual timeseries baselines to show an improvement over prior work or (2) further justification on why one would ever feel constrained to using a foundation model to interpret time-series data (i.e., how often or why text interpretation would be the only alternative), which is currently unclear to me.
2. The related work makes it unclear what has been done on (1) visual parsing of plots in general and (2) making use of the native visual capabilities of foundation models. The discussion of related work could use more description of trends in past work and how the current work improved upon them directly.
3. If the justification for why it is ok to have comparable performance on real-world tasks is that the vision encoder is more cost-efficient than the text encoder, more rigorous analysis of this improvement in cost efficiency seems necessary to show that the model provides any practical improvement in the real-world analysis.
4. The description of the method and tasks are very vague – there does not seem to be enough detail in the main text to understand what the method actually does, or what the tasks are measuring (what is being measured, how are they being evaluated, what is the significance of each task).

Other, more minor comments:
1. Table 2 is hard to parse – the visualization in Figure 1 seems to convey more information and more effectively (such as true value rather than just relative performance). Could you remove the table and use the space to describe the tasks in more detail?
2. Results across different noise/number of points rather than just aggregate performance for that task type would be interesting.
3. The real-world IMU data patterns seem considerably more complex than the pattern recognition tasks in the synthetic data. It would be interesting to see more description of how the results from synthetic and real-world tasks complement or relate to each other.

**Questions:**

1. What is the prevalence of plot vs. numerical text data in the typical available data? The benefit of this method seems dependent on currently having substantial, unused plotted time-series data.
2. Using the vision encoder to interpret visual input improves performance over text-only input on only some of the pairs of real-world example task and foundation model. Why does the method perform well or poorly on various tasks, and why is there so much variation in the performance of the method between different foundation models (e.g., Gemini Flash 1.5 vs GPT4o in Figure 3)?

---

> ### Author Response · Authors · 2024-11-27
> **Response to reviewer comments [1 / 2]**
>
> Thanks very much for your thorough reading and helpful feedback!
>
> **Regarding your specific questions:**
>
> > “What is the prevalence of plot vs. numerical text data in the typical available data? The benefit of this method seems dependent on currently having substantial, unused plotted time-series data. Using the vision encoder to interpret visual input improves performance over text-only input on only some of the pairs of real-world example task and foundation model.”
>
>
> As a point of clarification, our focus is not on ingesting previously unused plots of time-series data, but rather investigating for any time-series data whether a model can more easily reason about either the numeric or a (de novo) plotted representation of that same data.  With that in mind, we do not believe that the relative prevalence of plot versus text data is relevant, as we believe our argument holds for any time-series data.
>
>
> We acknowledge the second point that the visual performance is not strictly better than the text performance in all cases, but as shown in Table 2 on page 5 it is non-inferior in the vast majority of cases (39 out of 42 results), and comes with efficiency benefits over the text representation as described in Section 4.4 on page 9. Furthermore, in cases where the text approach performs better than the visual approach (e.g., the readiness task on Gemini Flash as described in Section 4.2 on page 9) the difference is small and within distributional overlap, so that the benefit of using the text approach is not strong.
>
> > “Why does the method perform well or poorly on various tasks, and why is there so much variation in the performance of the method between different foundation models (e.g., Gemini Flash 1.5 vs GPT4o in Figure 3)?”
>
>
> This is an excellent question, but unfortunately beyond the scope of our work as our focus is not on explainability. As the models we use are closed-source, we can speculate that the reason must be some mix of (unreleased) details of training data, model size, numerical tokenization mechanism and vision encoding mechanism. This question would form the basis of very interesting further work, which we have included in the final paragraph of the expanded Section 5 (Conclusions and Future Work) on page 10.
>
> **To your other points of feedback:**
>
> > “It seems like the actual task of recognizing the types of visual patterns achieved by this method are not inherently novel – Section 2 shows that models are already able to parse plots and tables. So the contribution of the paper would be more clear if the authors provided (1) comparisons to actual timeseries baselines to show an improvement over prior work or (2) further justification on why one would ever feel constrained to using a foundation model to interpret time-series data (i.e., how often or why text interpretation would be the only alternative), which is currently unclear to me.”
>
>
> The novelty of our work lies in the systematic investigation of the relative performances of visual versus textual encodings on these tasks across foundational models.  To be clear, we are not claiming that our approach replaces or obviates training task-specific models, which we expect will always achieve the highest absolute performance, but rather quantifies the differential ability of models to ingest visual versus textual information.  As such, the appropriate baseline is the performance of a foundational model on the text representation of the data.
>
>
> To our knowledge, this differential performance has not been rigorously studied in the literature, and is an important contribution to the field in the context of understanding and optimizing the use of foundation models.  Beyond the theoretical contribution of understanding model capabilities, our work also has important real-world implications.  Namely, as modern foundational models continue maturing and increasing their real-world adoption, users will expect the ability to interact with data-intensive products such as wellness trackers (Fitbit, Whoop, Apple Watch etc) via a natural language interface, which will necessitate a mechanism for foundation models likely to be incorporated into these products to best make sense of time-series data.  Current models such as Gemini Pro and GPT4o are already showing great progress in their multimodal capabilities, and this trend will likely continue such that text representations are no longer the default or optimum, especially for complex data types such as time-series as shown in our work.
>
>
> Thanks for raising these important points – we have revised the manuscript to include parts of our response here.  The point on the appropriate baseline has been added to the second paragraph of Section 3 (Methodology) starting on page 3, and the relevance has been added to the final paragraph of Section 1 (Introduction) on page 2 and the first bullet point of Section 5 (Conclusions and Future Work) on page 10.

---

> > ### Author Response · Authors · 2024-11-27
> > **Response to reviewer comments [2 / 2]**
> >
> > > “The related work makes it unclear what has been done on (1) visual parsing of plots in general and (2) making use of the native visual capabilities of foundation models. The discussion of related work could use more description of trends in past work and how the current work improved upon them directly.”
> >
> > We appreciate the feedback and have expanded on, and re-organised, the Related Works section on pages 2 and 3.  We also emphasise that our work is not claiming to improve any past work on plot representations, but rather studying the differential performance on existing vision versus text encodings of time-series data for various available models.
> >
> >
> > > “If the justification for why it is ok to have comparable performance on real-world tasks is that the vision encoder is more cost-efficient than the text encoder, more rigorous analysis of this improvement in cost efficiency seems necessary to show that the model provides any practical improvement in the real-world analysis.”
> >
> > We note first that the visual and text performances are comparable only in 1 real-world task (Readiness, on page 9), which we argue is in the low-data regime and unlikely to show clearer trends when plotted versus when presented as a textual table.  This result should also be taken in the context of all tasks we present as overall there is strong evidence of non-inferiority of vision over text across the tasks, as shown by 38 out of 42 results in Table 2 on page 5 having stronger plot performance on page 5 having stronger plot performance, so that if one were to apply a foundation model to a new time-series with unknown properties with limited resources to optimize responses (as might be the case in a user-facing product or a deployed API that might be presented with any sort of data), the visual approach on balance is likelier to yield better results; the cost benefit complements the increased performance but isn't itself necessarily the primary driver of real-world utility. Thanks for the question – we have included parts of the response in the final paragraph of Section 1 (Introduction) on page 2 and the first bullet point of Section 5 (Conclusions and Future Work) on page 10.
> >
> > > “The description of the method and tasks are very vague – there does not seem to be enough detail in the main text to understand what the method actually does, or what the tasks are measuring (what is being measured, how are they being evaluated, what is the significance of each task).”
> >
> > Thanks for pointing this out. We have included more details in Section 4.1 (Synthetic Data Tasks) on pages 6 and 7.  We have also expanded on the task-specific information available in the Supplementary Information Sections A.1.1 and A.1.2, which we have added as a forward reference in the first paragraph of Section 4.1 on page 6.
> >
> > > “Table 2 is hard to parse – the visualization in Figure 1 seems to convey more information and more effectively (such as true value rather than just relative performance). Could you remove the table and use the space to describe the tasks in more detail?”
> >
> > Thank you for the useful suggestion, we have reduced the number of rows in Table 2 and improved its readability on page 5.  We were also able to describe the tasks in more detail.
> >
> > > “Results across different noise/number of points rather than just aggregate performance for that task type would be interesting.”
> >
> > Thank you for raising this point, we have made it explicit in the first paragraph of the main text Section 4.1 (Synthetic Data Tasks) on page 6 that this information is available in the Supplementary Information Section A.2 (Further Results) starting on page 23.
> >
> > > “The real-world IMU data patterns seem considerably more complex than the pattern recognition tasks in the synthetic data. It would be interesting to see more description of how the results from synthetic and real-world tasks complement or relate to each other.”
> >
> > This is a good point - we have added language in the first paragraph of Section 4.2 (Real-world Tasks) on page 7 to connect the synthetic and real-world tasks better.
> >
> > We describe that the synthetic tasks build on complexity from single-step understanding of one trend globally (functional form identification) to understanding several trends simultaneously using global and local information (correlation and cluster counting) and complex multi-step reasoning (derivative identifications). Understanding the IMU traces requires simultaneous understanding of six trends (3 axes each for accelerometer and gyroscope), and requires complex reasoning to identify patterns in the few-shot examples and apply them to the inference tasks.  Furthermore, the two IMU tasks are different in that fall detection requires identifying a local spike, while activity detection requires understanding a global trend.  Hence, the reasoning required to make sense of the IMU traces can be seen as a combination of the individual reasonings we are probing with our synthetic tasks.

---

> > > ### Comment · Reviewer_V8rN · 2024-12-01
> > > **Thank you!**
> > >
> > > Thank you for taking the time to provide these detailed responses! While the responses were helpful, and I believe the changes mentioned will improve the clarity of the paper, my assessment of the paper's overall contribution and novelty remains the same—so I will retain my score. Thank you!

---

### Official Review · Reviewer_LLVU · 2024-11-05

**Soundness:** 2
**Presentation:** 2
**Contribution:** 4
**Rating:** 5
**Confidence:** 5

**Summary:**

This paper demonstrates that for some synthetic and real-world time series understanding problems (for example, detecting the functional form of a time series, and recognizing human activity), using the visual component of multimodal foundation models is better than using their text component. This is an interesting finding which can likely lead to better time series understanding performance, with fewer tokens.

**Strengths:**

1. The paper shows an interesting finding on a few synthetic and real-world tasks.
2. The authors conduct rigorous statistical experiments to evaluate differences between textual and visual presentation of time series inputs.

**Weaknesses:**

1. **Writing:** I would encourage the authors to spend some more time improving the writing of the manuscript. For example, there's a lot of forward references to the appendix and supplementary material with key information. This not only makes reading the manuscript harder, but the leaves the reader to wonder about some basic questions, for example how is the synthetic time series generated? The captions should be written so they communicate a story, rather than just the mechanics of the plot. The rationale behind some design decisions are not explained, for example the use of structured prompting. Also I would encourage the authors to use the space more effectively, some of the tables and figures can be made smaller, some figures and long descriptions of statistical tests can also be moved to the appendix. The extra space can then be used to have a more thorough related work section, for example covering how this work is positioned in the context of time series foundation models [1--4], multimodal time series and text models [5], how the are synthetic time series generated, why are some tasks chosen versus others, etc.
2. **Baselines:** Looking at the paper it is hard to judge whether the models are even doing well in these understanding tasks because only improvement of vision-based pipeline over the text-based pipeline is shown. Moreover, how do humans, simple time series models, other foundation models do? Only in Figure 4, do the authors mention "state-of-the-art", but fail to mention which model they are referring to. Moroever, the authors only use closed-source models, while open-source multimodal models are available. Including, results from these models are likely to aid reproducibility, and improve the rigour of the experiments. I would also encourage the authors to improve the readability of Table 1, for example I am not sure what "understanding of one overall trend" means, or what does "Num points" correspond to.
3. **Tasks:** The authors claim that they evaluate the "reasoning" capability of these models, but it is unclear how that is done? It's also unclear why some of these tasks were chosen, versus other tasks, or if reasoning elicitation techniques such as Chain-of-Thought or its variants were used. I would encourage the authors to review some recent work on evaluating reasoning in time series models [6] and LLMs [7]. These studies seem to have been made public very close to the ICLR deadline, but still present some useful and interesting results and experimental setups which might be of value in improving this manuscript.
4. **Plots:** Some questions about the nature of plots remain unanswered. What kinds of problems are benefited by visual processing. I am assuming problem which can be reasoned about by the shape of the time series. Also how were the plots generated? How does the plot generation mechanism affect the results?
5. **Cost:**: I like the cost argument, but one might also argue that the raw time series can be sub-sampled to answer the same questions. For a synthetic time series where the goal is to answer if something is periodic or not, the model doesn't need 1000 time points, and 100 uniformly sub-sampled points might suffice.

### References
1. Das, Abhimanyu, et al. "A decoder-only foundation model for time-series forecasting." arXiv preprint arXiv:2310.10688 (2023).
2. Woo, Gerald, et al. "Unified training of universal time series forecasting transformers." arXiv preprint arXiv:2402.02592 (2024).
3. Goswami, Mononito, et al. "Moment: A family of open time-series foundation models." arXiv preprint arXiv:2402.03885 (2024).
4. Ansari, Abdul Fatir, et al. "Chronos: Learning the language of time series." arXiv preprint arXiv:2403.07815 (2024).
5. Cai, Yifu, et al. "Jolt: Jointly learned representations of language and time-series." Deep Generative Models for Health Workshop NeurIPS 2023. 2023.
6. Potosnak, Willa, et al. "Implicit Reasoning in Deep Time Series Forecasting." arXiv preprint arXiv:2409.10840 (2024).
7. Cai, Yifu, et al. "TimeSeriesExam: A time series understanding exam." arXiv preprint arXiv:2410.14752 (2024).

**Questions:**

1. How are synthetic time series generated?
2. How are the options of the time series understanding questions generated?
3. What kinds of correlation between two lines are your measuring? Cross-correlation?

See other questions above.

---

> ### Author Response · Authors · 2024-11-27
> **Response to reviewer comments [1 / 3]**
>
> Thanks very much for your thorough reading and helpful feedback!
>
> **Regarding your specific questions:**
>
> > "How are synthetic time series generated?"
>
> We have added key information about the synthetic tasks that was previously in the Supplementary Information into Section 4.1 (Synthetic Data Tasks) of the main text starting on page 6, and have expanded the details for how the tasks were implemented in Supplementary Section A.1.1 (Synthetic Tasks) starting on page 13.
>
> > "How are the options of the time series understanding questions generated?"
>
> Also detailed in Supplementary Section A.1.1 (Synthetic Tasks) starting on page 13, all the synthetic time series understanding tasks are generated from a fixed range of values based on the task:
> - one of five functional forms for 'functional form identification'
> - a set of slope coefficients for each line for 'correlation of two lines'
> - a range of the number of distinct clusters for '2D cluster counting'
> - the derivative of one of the multiple choice options for 'derivative identification' and 'quadratic derivative identification'
>
> > "What kinds of correlation between two lines are you measuring? Cross-correlation?"
>
> For the correlation task we are taking the sign of the Pearson correlation coefficient between the two lines as the ground truth for whether the lines were positively or negatively correlated.
> In Supplementary Section A.4.2 (Correlation of two lines) we include the prompts showing how this was phrased to the model for the task.
> We have expanded the details in Section 4.1 (Synthetic Data Tasks) starting on page 6 to clarify this.
>
> **Regarding your further points and questions raised:**
>
> > "For example, there's a lot of forward references to the appendix and supplementary material with key information. This not only makes reading the manuscript harder, but the leaves the reader to wonder about some basic questions, for example how is the synthetic time series generated"
>
> We empathize with you on this issue, and found we had to carefully balance the amount of details on the methods in the main, page-limited manuscript with the presentation of primary results. As the focus of the paper is the relative performance differences in time series understanding between plot and text based methods, and we explored this over a range of synthetic and real-world tasks we felt that including the high level elements of each task and covering the range of findings to demonstrate the higher performance of plot methods was the most important.
>
> The specific details of the synthetic time series tasks and the generation of their datasets formed the first 3 pages of the Supplementary Information. We agree they form an important part of the research, and wanted to ensure they were covered in necessary detail, but felt that promoting all of that detail to the main manuscript would result in other key elements needing to be removed. However, as above, we expanded the details in Section 4.1 (Synthetic Data Tasks) starting on page 6 to include more of the key information in the main text. Similarly the details on the range of ablation experiments performed were also important, but on balance the bulk of this was also placed in the Supplementary Information.
>
> > "The captions should be written so they communicate a story, rather than just the mechanics of the plot."
>
> Thank you for the suggestion; we have adjusted figure and table captions throughout to better convey the narrative of the paper.
>
> > "The rationale behind some design decisions are not explained, for example the use of structured prompting"
>
> For structured prompting specifically, the rationale was to more easily scale across different model APIs by using the response structuring layer as a mechanism to enable general code and enforce model response types / values.  We chose LangFun as our structured prompting library as it is open-source, and we include exact prompt templates in Supplementary Information Section A.4 starting on page 43 in the spirit of reproducibility.
>
> If there are further design decisions you feel need further clarification, we'll be happy to do so.
>
>
> > "some of the tables and figures can be made smaller, some figures and long descriptions of statistical tests can also be moved to the appendix"
>
> We have made Table 2 and Figures 2-4 smaller, which allowed us to expand on the text while remaining in the page limit; thank you for the suggestion.
>
> > "more thorough related work section, for example covering how this work is positioned in the context of time series foundation models [1--4], multimodal time series and text models [5]"
>
> We appreciate the feedback and have expanded the Related Work section in pages 2-3 to better position our work in the context of existing work on time series foundation models and multimodal time series and text models. In particular, the references you very helpfully provided have been included in the “Time-series models” subsection on page 2.

---

> > ### Author Response · Authors · 2024-11-27
> > **Response to reviewer comments [2 / 3]**
> >
> > > "Baselines: Looking at the paper it is hard to judge whether the models are even doing well in these understanding tasks because only improvement of vision-based pipeline over the text-based pipeline is shown. Moreover, how do humans, simple time series models, other foundation models do? Only in Figure 4, do the authors mention "state-of-the-art", but fail to mention which model they are referring to."
> >
> > We emphasise that our work is the systematic investigation of the relative performances of visual versus textual encodings on these tasks across foundational models, rather than absolute performance on any specific task for a given modality of input data.  As such, the appropriate baseline, and the one which we employ, is the performance of a foundational model on the text representation of the data. We have clarified this in the last paragraph to start on page 3 in Section 3 (Methodology).
> >
> > To put the absolute performances into context, we also provide random baselines to show if there is utility at all in using a foundation model in Figures 1-5.
> >
> > For the real-world fall detection task we include the state-of-the-art performance reported by Aziz et al. (2017). This is presented in the caption of the Figure 3 on page 8, since their reported metrics of sensitivity and specificity don't match what we're showing in the figure, but we also include direct comparisons of sensitivity and specificity in Supplementary Table S1 on page 20.
> > For the activity recognition task we include the state-of-the-art performance reported by Kumar & Selvam (2022) directly in Figure 4 on page 8.
> >
> > > "...the authors only use closed-source models, while open-source multimodal models are available. Including, results from these models are likely to aid reproducibility, and improve the rigour of the experiments."
> >
> > We recognise the importance and availability of open-source multimodal models. TimeSeriesExam: A Time Series Understanding Exam (Cai et al) suggests that currently “closed source models such as GPT-4 and Gemini understand time series concepts significantly better than their open-source counterparts.” As we are showing the difference in understanding between the data being presented as a visual plot rather than as raw text, using more capable models provides us more scope to be able demonstrate this. Even though the models we used are commercial, closed-source models, they are still publicly available to the research community, albeit at a cost. We ensured to include the exact model version numbers used to aid in reproducibility against these models.
> >
> > We explored using [PaliGemma](https://arxiv.org/pdf/2407.07726) as an additional model during the research, but various current limitations (ordering of inputs, single image input, level of instruction tuning) made it impractical. As open-source multimodal models continue to improve we expect to be able to demonstrate similar results to what we've shown on closed-source models in the future.
> >
> > > "I would also encourage the authors to improve the readability of Table 1, for example I am not sure what "understanding of one overall trend" means, or what does "Num points" correspond to."
> >
> > Thank you for pointing this out; we have revised Table 1 on page 5 to be more readable, in particular renaming the “Num points” column to “Time-series length” and expanding on what each column means in the caption.
> >
> > To your specific points, the "understanding of one overall trend" is a distillation of the existing functional form identification description "This is the simplest task that requires only identifying one overall trend and correctly classifying it into one of five functional tasks (linear, quadratic, cubic, exponential or periodic)" (from page 6) into the key element of "reasoning" that the task is attempting to assess.
> >
> >
> > > "The authors claim that they evaluate the "reasoning" capability of these models, but it is unclear how that is done? It's also unclear why some of these tasks were chosen, versus other tasks, or if reasoning elicitation techniques such as Chain-of-Thought or its variants were used"
> >
> > Thanks for raising this important point.  We chose a set of synthetic tasks that we hypothesised would represent different steps in human reasoning it would take to get to the right answer, though we didn't interrogate the actual reasoning that the models may have used to come up with their answers.
> >
> > We weren't referring to any specific modelling capability or technique, and especially we don't mean to conflate it with formal reasoning approaches such as Chain-of-Thought. We have updated the manuscript to make this distinction clear in the first paragraph of Section 3 (Methodology) on page 3, and we have expanded on the connection between the synthetic and real-world tasks we have chosen in the first paragraph of Section 4.2 (Real-world Tasks) on page 7.

---

> > > ### Author Response · Authors · 2024-11-27
> > > **Response to reviewer comments [3 / 3]**
> > >
> > > > "I would encourage the authors to review some recent work on evaluating reasoning in time series models [6] and LLMs [7] These studies seem to have been made public very close to the ICLR deadline, but still present some useful and interesting results and experimental setups which might be of value in improving this manuscript."
> > >
> > > Thank you very much for bringing these recent papers to our attention. We have included these in the expanded Related Works section, specifically in the “Time-series Models” and “Measuring Understanding” subsections on pages 2 and 3 respectively.
> > >
> > > > "Some questions about the nature of plots remain unanswered. What kinds of problems are benefited by visual processing. I am assuming problem which can be reasoned about by the shape of the time series. Also how were the plots generated? How does the plot generation mechanism affect the results?"
> > >
> > > We ran ablations on the plot generations over resolution, figure size, plot style, marker types and sizes, color palettes and plot components (in Section 4.3 on page 9 and further details in Supplementary Information Section A.3 starting on page 28) and found that there wasn't a strong effect on the results. We have added Supplementary Figure S14 on page 29 to help visualise the range of plot generation options covered in these ablations.
> > >
> > > We have updated the manuscript to add more details for how the plots were generated by including plotting code snippets in Supplementary Information Section A.1 (Detailed Task Descriptions) starting on page 13.
> > >
> > > We didn't robustly investigate the set of real world time series problems that are benefited by visual processing. We found that the IMU based problems were, but the Readiness task was not. The nature of the time series data and the real world task itself are obviously key and we hypothesise that the benefit is more obvious when the amount of data exceeds what's reasonably presentable in a text table, but investigating this thoroughly is an area of future research.
> > >
> > > > "I like the cost argument, but one might also argue that the raw time series can be sub-sampled to answer the same questions. For a synthetic time series where the goal is to answer if something is periodic or not, the model doesn't need 1000 time points, and 100 uniformly sub-sampled points might suffice."
> > >
> > > This is a very valid point and thanks for raising it.
> > >
> > > For the synthetic data tasks, the performance on a lower number of points for the same tasks is a proxy for sub-sampling the full time-series.  As shown in the "Results as a function of number of points" figures in Supplementary Figures S9-S13 starting on page 24, this intuition holds and shows that performance on these tasks can be comparable if the data is sub-sampled.
> > >
> > > For real-world tasks however this is not necessarily the case, as depending on the task there is likely to be a sample rate below which the signal is no longer present for the model to discern. In response to your point I ran some additional experiments on the Fall Detection task, and subsampling below the original 10Hz rate further decreases the performance of the text method. So while it does become cheaper, it becomes less and less accurate. The trend wasn't exactly the same for activity recognition, where performance reached a plateau with further subsampling but remained below the performance of the plot performance. So in a way the plotting of the time-series data can be analogous to a "compression" of the full sample rate data to a fixed token size and therefore a fixed cost, whereas the textual representation can be "compressed" to a variable number of tokens and therefore potentially a variable cost and variable performance. Though with the plot method, there are still choices that can be made to vary the cost. For example there was a marginal gain observed in the Activity Recognition task if the 6 IMU axes were split and the accelerometer and gyroscope axes plotted independently (new Supplementary Figure S8 on page 22) albeit with an additional cost for the additional image tokens. In a real world usage, these cost-performance tradeoffs would need to be considered in the context of the application.
> > >
> > > We have updated the manuscript to discuss sub-sampling as a strategy to reduce costs for the text method in the penultimate paragraph of Section 4.4 (Cost) on page 9.

---

> > > > ### Comment · Reviewer_LLVU · 2024-11-30
> > > > **Thanks for your rebuttal!**
> > > >
> > > > Dear Authors,
> > > >
> > > > Thank you so much for your detailed response and hard work, and for making significant changes to the manuscript during the rebuttal.
> > > >
> > > > For future reference, I would highly recommend highlighting changes in a different ink so it is easier to track them in the updated manuscript.
> > > >
> > > > I have updated my score based on the changes to the paper to reflect my current assessment of the paper.
> > > >
> > > > Good luck!

---

### Author Response · Authors · 2024-11-27
**General response to reviews**

We thank all the reviewers for their time spent reviewing our submissions and their detailed feedback, and look forward to discussing further details here.

We are happy that the reviewers noted that our approach is “creative” (fw6B), our experimental methodology is “rigorous” (LLVU) and that overall they “believe it add[s] significant value to the ICLR community” (RmeP).

A few common themes emerged in the reviews. We respond to each reviewer individually where questions arise in the comments, and we summarise here the main points:
- **Novelty of our work**: We provide the first (to our knowledge) rigorous evaluation of multimodal foundation model performances on time series understanding tasks when presented the same data either visually (as plots) or textually (as strings of numbers). We are not claiming to contribute the idea of visually representing time-series data, nor claiming a new method in doing so, and instead focussing on demonstrating that modern foundation models perform better on time-series understanding tasks when presented the data as plots rather than as numerical lists.
- **Relevance of our work**: Generally, our empirical results further the field’s understanding of the capabilities of modern foundation models to interpret time-series via different modalities. From a real-world perspective, the increasing power, and user adoption, of the multimodal capabilities of consumer-facing models (Gemini Pro, GPT-4o, etc) imply text-only representations are becoming increasingly less important for complex data, of which time-series is a prime example. In 38 out of 42 results we present in Table 1 the visual approach is better than the text approach, implying that on balance, plotting time series is more likely to yield better, cheaper, results. This is important especially in situations (such as user-facing chatbots) where it might not be known a priori which kinds of data will be presented to the model.
- **Appropriate baselines**: As we are studying the differential performance of plot versus text approaches for the same model on a given task, rather than studying the absolute performance of either approach, the relevant baseline is the text performance of the models we investigate. We also are not claiming that our approach replaces or obviates training task-specific models, which we expect will continue to achieve the highest absolute performance. We included state-of-the-art baselines of purpose-built models against two of our real world tasks results to put them in context, though direct comparison with them is not the main point of the paper.
- **Details of tasks**: Several reviewers mentioned that the text could benefit by making task details clearer in the main text of the paper, as we had originally included some details in the Supplementary Information.

We provide here a summary of the changes we’ve made in the manuscript to reflect reviewer feedback:
- The above points have been highlighted throughout, especially in the introduction and conclusion.
- The related work section has been re-organised into approaches and expanded.
- More details about our synthetic experiments have been moved into the main text, and additional information and code snippets for the tasks have been added to the Supplementary Information.
- Figure and table captions have been modified to contain conclusions that can be drawn.
- The future work section has been expanded.

Thank you to all the reviewers again for helping make our work clearer and more impactful.  We are very grateful for the feedback, and would be happy to respond to any further points.

---

### Meta-Review · Area_Chair_DcXV · 2024-12-20

**Metareview:**

This paper has been evaluated by 4 knowledgeable reviewers and their initial opinions varied: 1 strong rejection, 1 marginal rejection and 1 marginal acceptance. The authors provided extensive rebuttals but that did not help the scores. Even though the key concept presented in the paper is relevant to ICLR and sufficiently well motivated, reviewers pointed out unjustified claims, missing important baselines and relevant literature, and some questioned the novelty of the core idea.

**Additional Comments On Reviewer Discussion:**

The reviewers engaged in a discussion to align their opinions, and that discussion has reduced variance of the scores and reinforced the consensual opinion that this work is not ready for ICLR in its current form.

---

### Decision · Program_Chairs · 2025-01-22

Reject